# Near-Optimal Regret Bounds for Model-Free RL in Non-Stationary Episodic MDPs

## Abstract

We consider model-free reinforcement learning (RL) in non-stationary Markov decision processes (MDPs). Both the reward functions and the state transition distributions are allowed to vary over time, either gradually or abruptly, as long as their cumulative variation magnitude does not exceed certain budgets. We propose an algorithm, named Restarted Q-Learning with Upper Confidence Bounds (RestartQ-UCB), for this setting, which adopts a simple restarting strategy and an extra optimism term. Our algorithm outperforms the state-of-the-art (model-based) solution in terms of dynamic regret. Specifically, RestartQ-UCB with Freedman-type bonus terms achieves a dynamic regret of $\widetilde{O}(S^{\frac{1}{3}} A^{\frac{1}{3}} \Delta^{\frac{1}{3}} H T^{\frac{2}{3}})$, where $S$ and $A$ are the numbers of states and actions, respectively, $\Delta > 0$ is the variation budget, $H$ is the number of steps per episode, and $T$ is the total number of steps. We further show that our algorithm is near-optimal by establishing an information-theoretical lower bound of $\Omega(S^{\frac{1}{3}} A^{\frac{1}{3}} \Delta^{\frac{1}{3}} H^{\frac{2}{3}} T^{\frac{2}{3}})$, which to the best of our knowledge is the first impossibility result in non-stationary RL in general.

## 1 Introduction

Reinforcement learning (RL) studies the class of problems where an agent maximizes its cumulative reward through sequential interaction with an unknown but fixed environment, usually modeled by a Markov Decision Process (MDP). At each time step, the agent takes an action, receives a random reward drawn from a reward function, and then the environment transitions to a new state according to an unknown transition kernel. In classical RL problems, the transition kernel and the reward functions are assumed to be time-invariant. This stationary model, however, cannot capture the phenomenon that in many real-world decision-making problems, the environment, including both the transition dynamics and the reward functions, is inherently evolving over time. Non-stationarity exists in a wide range of applications, including online advertisement auctions (Cai et al., 2017; Lu et al., 2019), dynamic pricing (Board, 2008; Chawla et al., 2016), traffic management (Chen et al., 2020), healthcare operations (Shortreed et al., 2011), and inventory control (Agrawal & Jia, 2019).

Among the many intriguing applications, we specifically emphasize two research areas that can significantly benefit from progress on non-stationary RL, yet their connections have been largely overlooked in the literature. The first one is sequential transfer in RL (Tirinzoni et al., 2020) or multi-task RL Brunskill & Li (2013). In this setting, the agent encounters a sequence of tasks over time with different system dynamics and reward functions, and seeks to bootstrap learning by transferring knowledge from previously-solved tasks. The second one is multi-agent reinforcement learning (MARL) (Littman, 1994), where a set of agents collaborate or compete in a shared environment. In MARL, since the transition and reward functions of the agents are coupled, the environment is non-stationary from each agent's own perspective, especially when the agents learn and update policies simultaneously. A more detailed discussion on how non-stationary RL can benefit sequential transfer, multi-task, and multi-agent RL is given in Appendix A.

Learning in a non-stationary MDP is highly non-trivial due to the following challenges. The first one is the *exploration vs. exploitation* challenge inherited from standard (stationary) RL. An agent needs to explore the uncertain environment efficiently while maximizing its rewards along the way. Classical solutions in stationary RL oftentimes leverage the "optimism in the face of uncertain" principle that adopts an upper confidence bound to guide exploration. These bounds can be either an optimistic estimate of the state transition distributions in model-based solutions (Jaksch et al., 2010),

| Setting | Algorithm | Regret | Model-free? | Comment |
|---|---|---|---|---|
| Undis-counted | Jaksch et al. (2010) | $\widetilde{O}(S\ A^{\frac{1}{2}}L^{\frac{1}{3}}D\ T^{\frac{2}{3}})$ | ✗ | only abrupt changes |
| | Gajane et al. (2018) | $\widetilde{O}(S^{\frac{2}{3}}A^{\frac{1}{3}}L^{\frac{1}{3}}D^{\frac{2}{3}}T^{\frac{2}{3}})$ | ✗ | only abrupt changes |
| | Ortner et al. (2019) | $\widetilde{O}(S\ A^{\frac{1}{2}}\Delta^{\frac{1}{3}}D\ T^{\frac{2}{3}})$ | ✗ | requires local budgets |
| | Cheung et al. (2020) | $\widetilde{O}(S^{\frac{2}{3}}A^{\frac{1}{2}}\Delta^{\frac{1}{4}}D\ T^{\frac{3}{4}})$ | ✗ | does not require $\Delta$ |
| | Lower bound | $\Omega(S^{\frac{1}{3}}A^{\frac{1}{3}}\Delta^{\frac{1}{3}}D^{\frac{2}{3}}T^{\frac{2}{3}})$ | | |
| Episodic | Domingues et al. (2020) | $\widetilde{O}(S\ A^{\frac{1}{2}}\Delta^{\frac{1}{3}}H^{\frac{4}{3}}T^{\frac{2}{3}})$ | ✗ | also metric spaces |
| | RestartQ-UCB* | $\widetilde{O}(S^{\frac{1}{3}}A^{\frac{1}{3}}\Delta^{\frac{1}{3}}H\ T^{\frac{2}{3}})$ | ✓ | |
| | Lower bound* | $\Omega(S^{\frac{1}{3}}A^{\frac{1}{3}}\Delta^{\frac{1}{3}}H^{\frac{2}{3}}T^{\frac{2}{3}})$ | | |

Table 1: Dynamic regret comparisons for RL in non-stationary MDPs. $S$ and $A$ are the numbers of states and actions, $L$ is the number of abrupt changes, $D$ is the maximum diameter, $H$ is the number of steps per episode, and $T$ is the total number of steps. Gray cells denote results from this paper.

or an optimistic estimate of the $Q$-values in the model-free ones (Jin et al., 2018; Zhang et al., 2020). An additional challenge in non-stationary RL is the trade-off between *remembering and forgetting*. Since the system dynamics vary from one episode to another, all the information collected from previous interactions are essentially out-of-date and biased. In fact, it has been shown that a standard RL algorithm might incur a linear regret if the non-stationarity is not handled properly (Ortner et al., 2019). On the other hand, the agent does need to maintain a sufficient amount of information from history for future decision making, and *learning what to remember* becomes a further challenge.

In this paper, we introduce an algorithm, named *Restarted Q-Learning with Upper Confidence Bounds* (RestartQ-UCB), to address the aforementioned challenges in non-stationary RL. Our algorithm utilizes an extra optimism term for exploration, in addition to the standard Hoeffding/Bernstein-based bonus in the upper confidence bound, to counteract the non-stationarity of the MDP. This additional bonus term guarantees that our optimistic $Q$-value is still an upper bound of the optimal $Q^\star$-value even when the environment changes. To address the second challenge, we adopt a simple but effective restarting strategy that resets the memory of the agent according to a calculated schedule. Similar strategies have also been considered in non-stationary bandits (Besbes et al., 2014) and non-stationary RL in the un-discounted setting (Jaksch et al., 2010; Ortner et al., 2019). The restarting strategy ensures that our algorithm only refers to the most up-to-date experience for decision-making. A further advantage of our algorithm is that RestartQ-UCB is *model-free*. Compared with model-based solutions, our model-free algorithm is more time- and space-efficient, flexible to use, and more compatible with the design of modern deep RL architectures.

**Related Work.** Dynamic regret of non-stationary RL has been mostly studied using *model-based* solutions. Jaksch et al. (2010) consider the setting where the MDP is allowed to change abruptly $L$ times, and achieve a regret of $\widetilde{O}(SA^{\frac{1}{2}}L^{\frac{1}{3}}DT^{\frac{2}{3}})$, where $D$ is the maximum diameter of the MDP. A sliding window approach is proposed in Gajane et al. (2018) under the same setting. Ortner et al. (2019) generalize the previous setting by allowing the MDP to vary either abruptly or gradually at every step, subject to a total variation budget of $\Delta$. Cheung et al. (2020) consider the same setting and develop a sliding window algorithm with confidence widening. The authors also introduce a Bandit-over-RL technique that adaptively tunes the algorithm without knowing the variation budget. In a setting most similar to ours, Domingues et al. (2020) investigate non-stationary RL in the episodic setting. They propose a kernel-based approach when the state-action set forms a metric space, and their results can be reduced to an $\widetilde{O}(SA^{\frac{1}{2}}\Delta^{\frac{1}{3}}H^{\frac{4}{3}}T^{\frac{2}{3}})$ regret in the tabular case. Fei et al. (2020) also consider the episodic setting, but they assume stationary transition kernels and adversarial (subject to some smoothness assumptions) full-information rewards. The authors propose two

---

*Connections to stationary RL: Results in Table 1 hold for $\Delta > 0$. To derive an upper bound for $\Delta = 0$, we only need a simple modification in the proof of Theorem 3 by setting the number of epochs to be 1. This leads to an upper bound of $\widetilde{O}(H\sqrt{SAT})$, which matches the results given in Zhang et al. (2020). A similar modification in the proof of Theorem 4 results in a lower bound of $\Omega(H\sqrt{SAT})$ when $\Delta = 0$.

policy optimization algorithms, which are also the only model-free solutions that we are aware of in non-stationary RL. In contrast, we allow both the transition kernel and the reward function to change over time, and deal with bandit-feedback, which makes the setting in Fei et al. (2020) not directly comparable. Table 1 compares our regret bounds with existing results that tackle the same setting as ours. Interested readers are referred to Padakandla (2020) for a comprehensive survey on RL in non-stationary environments. We would also like to mention another related line of research that studies online/adversarial MDPs (Yu & Mannor, 2009; Neu et al., 2010; Arora et al., 2012; Yadkori et al., 2013; Dick et al., 2014; Wang et al., 2018; Lykouris et al., 2019; Jin et al., 2019), but they mostly only allow variations in reward functions, and use static regret as performance metric. Finally, RL with low switching cost (Bai et al., 2019) also shares a similar spirit as our restarting strategy since it also periodically forgets previous experiences. However, such algorithms do not address the non-stationarity of the environment explicitly, and it is non-trivial to analyze its dynamic regret in terms of the variation budget.

Non-stationarity has also been considered in bandit problems. Under different non-stationary multi-armed bandit (MAB) settings, various methods have been proposed, including decaying memory and sliding windows (Garivier & Moulines, 2011; Keskin & Zeevi, 2017), as well as restart-based strategies (Auer et al., 2002; Besbes et al., 2014; Allesiardo et al., 2017). These methods largely inspired later research in non-stationary RL. A more recent line of work developed methods that do not require prior knowledge of the variation budget (Karnin & Anava, 2016; Cheung et al., 2019a) or the number of abrupt changes (Auer et al., 2019). Other related settings considered in the literature include Markovian bandits (Tekin & Liu, 2010; Ma, 2018), non-stationary contextual bandits (Luo et al., 2018; Chen et al., 2019), linear bandits (Cheung et al., 2019b; Zhao et al., 2020), continuous-armed bandits (Mao et al., 2020), and bandits with slowly changing rewards (Besbes et al., 2019).

**Contributions.** First, we propose RestartQ-UCB, the first model-free RL algorithm in the general setting of non-stationary MDPs, where both the transition kernel and reward functions are allowed to vary over time. Second, we provide dynamic regret analysis for RestartQ-UCB, and show that it outperforms even the model-based state-of-the-art solution. Third, we establish the first lower bounds in non-stationary RL, which suggest that our algorithm is optimal in all parameter dependences except for an $H^{\frac{1}{3}}$ factor, where $H$ is the episode length.

In the main text of this paper, we will present and analyze a simpler version of RestartQ-UCB with a Hoeffding-style bonus term. Replacing the Hoeffding term with a Freedman-style one will lead to a tighter regret bound, but the analysis is more involved. For clarity of presentation, we defer the exposition and analysis of the Freedman-based algorithm to the appendices. All missing proofs in the paper can also be found in the appendices.

## 2 PRELIMINARIES

**Model:** We consider an episodic RL setting where an agent interacts with a non-stationary MDP for $M$ episodes, with each episode containing $H$ steps. We use a pair of integers $(m, h)$ as a *time index* to denote the $h$-th step of the $m$-th episode. The environment can be denoted by a tuple $(\mathcal{S}, \mathcal{A}, H, P, r)$, where $\mathcal{S}$ is the finite set of states with $|\mathcal{S}| = S$, $\mathcal{A}$ is the finite set of actions with $|\mathcal{A}| = A$, $H$ is the number of steps in one episode, $P = \{P_h^m\}_{m \in [M], h \in [H]}$ is the set of transition kernels, and $r = \{r_h^m\}_{m \in [M], h \in [H]}$ is the set of mean reward functions. Specifically, when the agent takes action $a_h^m \in \mathcal{A}$ in state $s_h^m \in \mathcal{S}$ at the time $(m, h)$, it will receive a random reward $R_h^m(s_h^m, a_h^m) \in [0, 1]$ with expected value $r_h^m(s_h^m, a_h^m)$, and the environment transitions to a next state $s_{h+1}^m$ following the distribution $P_h^m(\cdot \mid s_h^m, a_h^m)$. It is worth emphasizing that the transition kernel and the mean reward function depend both on $m$ and $h$, and hence the environment is non-stationary over time. The episode ends when $s_{H+1}^m$ is reached. We further denote $T = MH$ as the total number of steps.

A deterministic policy $\pi : [M] \times [H] \times \mathcal{S} \to \mathcal{A}$ is a mapping from the time index and state space to the action space, and we let $\pi_h^m(s)$ denote the action chosen in state $s$ at time $(m, h)$. Define $V_h^{m, \pi} : \mathcal{S} \to \mathbb{R}$ to be the value function under policy $\pi$ at time $(m, h)$, i.e.,

$$V_h^{m, \pi}(s) \overset{\text{def}}{=} \mathbb{E}\left[ \sum_{h'=h}^{H} r_{h'}^m \left(s_{h'}, \pi_{h'}^m(s_{h'})\right) \mid s_h = s \right], s_{h'+1} \sim P_{h'}^m\left(\cdot \mid s_{h'}, a_{h'}\right).$$

Accordingly, the state-action value function $Q_h^{m,\pi} : \mathcal{S} \times \mathcal{A} \to \mathbb{R}$ is defined as:

$$Q_h^{m,\pi}(s,a) \stackrel{\text{def}}{=} r_h^m(s,a) + \mathbb{E}\left[\sum_{h'=h+1}^H r_{h'}^m(s_{h'}, \pi_{h'}^m(s_{h'})) \mid s_h = s, a_h = a\right].$$

For simplicity of notation, we let $P_h^m V_{h+1}(s,a) \stackrel{\text{def}}{=} \mathbb{E}_{s' \sim P_h^m(\cdot|s,a)}[V_{h+1}(s')]$. Then, the Bellman equation gives $V_h^{m,\pi}(s) = Q_h^{m,\pi}(s, \pi_h^m(s))$ and $Q_h^{m,\pi}(s,a) = (r_h^m + P_h^m V_{h+1}^{m,\pi})(s,a)$, and we also have $V_{H+1}^{m,\pi}(s) = 0, \forall s \in \mathcal{S}$ by definition. Since the state space, the action space, and the length of each episode are all finite, there always exists an optimal policy $\pi^\star$ that gives the optimal value $V_h^{m,\star}(s) \stackrel{\text{def}}{=} V_h^{m,\pi^\star}(s) = \sup_\pi V_h^{m,\pi}(s), \forall s \in \mathcal{S}, m \in [M], h \in [H]$. From the Bellman optimality equation, we have $V_h^{m,\star}(s) = \max_{a \in \mathcal{A}} Q_h^{m,\star}(s,a)$, where $Q_h^{m,\star}(s,a) \stackrel{\text{def}}{=} (r_h^m + P_h^m V_{h+1}^{m,\star})(s,a)$, and $V_{H+1}^{m,\star}(s) = 0, \forall s \in \mathcal{S}$.

**Dynamic Regret:** The agent aims to maximize the cumulative expected reward over the entire $M$ episodes, by adopting some policy $\pi$. We measure the optimality of the policy $\pi$ in terms of its *dynamic regret* (Cheung et al., 2020; Domingues et al., 2020), which compares the agent's policy with the optimal policy of each individual episode in the hindsight:

$$\mathcal{R}(\pi, M) \stackrel{\text{def}}{=} \sum_{m=1}^M \left(V_1^{m,\star}(s_1^m) - V_1^{m,\pi}(s_1^m)\right),$$

where the initial state $s_1^m$ of each episode is chosen by an adversary (and more specifically, by an oblivious adversary (Zhang et al., 2020)). Dynamic regret is a stronger measure than the standard (static) regret, which only considers the single policy that is optimal over all episodes combined.

**Variation:** We measure the non-stationarity of the MDP in terms of its *variation* in the mean reward function and transition kernels:

$$\Delta_r \stackrel{\text{def}}{=} \sum_{m=1}^{M-1} \sum_{h=1}^H \sup_{s,a} |r_h^m(s,a) - r_h^{m+1}(s,a)|, \quad \Delta_p \stackrel{\text{def}}{=} \sum_{m=1}^{M-1} \sum_{h=1}^H \sup_{s,a} \left\|P_h^m(\cdot \mid s,a) - P_h^{m+1}(\cdot \mid s,a)\right\|_1,$$

where $\|\cdot\|_1$ is the $L^1$-norm. Note that our definition of variation only imposes restrictions on the summation of non-stationarity across two different episodes, and does not put any restriction on the difference between two consecutive steps in the same episode; that is, $P_h^m(\cdot \mid s,a)$ and $P_{h+1}^m(\cdot \mid s,a)$ are allowed to be arbitrarily different. We further let $\Delta = \Delta_r + \Delta_p$, and assume $\Delta > 0$.

## 3   ALGORITHM: RESTARTQ-UCB

We present our algorithm Restarted Q-Learning with Hoeffding Upper Confidence Bounds (RestartQ-UCB Hoeffding) in Algorithm 1. Replacing the Hoeffding-style upper confidence bound in Algorithm 1 with a Freedman-style one will lead to a tighter regret bound, but for clarity of exposition, the latter version will be deferred to Algorithm 2 in Appendix C.

RestartQ-UCB breaks the $M$ episodes into $D$ *epochs*, with each epoch containing $K = \lceil \frac{M}{D} \rceil$ episodes (except for the last epoch which possibly has less than $K$ episodes). The optimal value of $D$ (and hence $K$) will be specified later in our analysis. RestartQ-UCB periodically restarts a Q-learning algorithm with UCB exploration at the beginning of each epoch, thereby addressing the non-stationarity of the environment. For each $d \in [D]$, define $\Delta_r^{(d)}$ to be the variation of the mean reward function within epoch $d$. By definition, we have $\sum_{d=1}^D \Delta_r^{(d)} \leq \Delta_r$. Further, for each $d \in [D]$ and $h \in [H]$, define $\Delta_{r,h}^{(d)}$ to be the variation of the mean reward at step $h$ in epoch $d$, i.e., $\Delta_{r,h}^{(d)} \stackrel{\text{def}}{=} \sum_{m=(d-1)K+1}^{\min\{dK,M\}-1} \sup_{s,a} \left|r_h^m(s,a) - r_h^{m+1}(s,a)\right|$. It also holds that $\sum_{h=1}^H \Delta_{r,h}^{(d)} = \Delta_r^{(d)}$ by definition. Define $\Delta_p^{(d)}$ and $\Delta_{p,h}^{(d)}$ analogously.

Since our algorithm essentially invokes the same procedure for every epoch, in the following, we focus our analysis on what happens inside one epoch only (and without loss of generality, we focus on epoch 1, which contains episodes $1, 2, \ldots, K$). At the end of our analysis, we will merge the results across all epochs.

---

**Algorithm 1:** RestartQ-UCB (Hoeffding)

1    **for** *epoch* $d \leftarrow 1$ *to* $D$ **do**
2      **Initialize:** $V_h(s) \leftarrow H - h + 1, Q_h(s,a) \leftarrow H - h + 1, N_h(s,a) \leftarrow 0, \check{N}_h(s,a) \leftarrow 0,$
      $\check{r}_h(s,a) \leftarrow 0, \check{v}_h(s,a) \leftarrow 0$, for all $(s,a,h) \in \mathcal{S} \times \mathcal{A} \times [H]$;
3      **for** *episode* $k \leftarrow (d-1)K + 1$ *to* $\min\{dK, M\}$ **do**
4        observe $s_1^k$;
5        **for** *step* $h \leftarrow 1$ *to* $H$ **do**
6          Take action $a_h^k \leftarrow \arg\max_a Q_h(s_h^k, a)$, receive $R_h^k(s_h^k, a_h^k)$, and observe $s_{h+1}^k$;
7          $\check{r}_h(s_h^k, a_h^k) \leftarrow \check{r}_h(s_h^k, a_h^k) + R_h^k(s_h^k, a_h^k), \check{v}_h(s_h^k, a_h^k) \leftarrow \check{v}_h(s_h^k, a_h^k) + V_{h+1}(s_{h+1}^k)$;
8          $N_h(s_h^k, a_h^k) \leftarrow N_h(s_h^k, a_h^k) + 1, \check{N}_h(s_h^k, a_h^k) \leftarrow \check{N}_h(s_h^k, a_h^k) + 1$;
9          **if** $N_h(s_h^k, a_h^k) \in \mathcal{L}$    // Reaching the end of the stage
10         **then**
11           $b_h^k \leftarrow \sqrt{\frac{H^2}{\check{N}_h(s_h^k, a_h^k)}\iota} + \sqrt{\frac{1}{\check{N}_h(s_h^k, a_h^k)}\iota}, \; b_\Delta \leftarrow \Delta_r^{(d)} + H\Delta_p^{(d)}$;
12           $Q_h(s_h^k, a_h^k) \leftarrow \min\left\{ \frac{\check{r}_h(s_h^k, a_h^k)}{\check{N}_h(s_h^k, a_h^k)} + \frac{\check{v}_h(s_h^k, a_h^k)}{\check{N}_h(s_h^k, a_h^k)} + b_h^k + 2b_\Delta, Q_h(s_h^k, a_h^k) \right\}$;     $(*)$
13           $V_h(s_h^k) \leftarrow \max_a Q_h(s_h^k, a)$;
14           $\check{N}_h(s_h^k, a_h^k) \leftarrow 0, \check{r}_h(s_h^k, a_h^k) \leftarrow 0, \check{v}_h(s_h^k, a_h^k) \leftarrow 0$;

---

For each triple $(s, a, h) \in \mathcal{S} \times \mathcal{A} \times [H]$, we divide the visitations (within epoch 1) to the triple into multiple *stages*, where the length of the stages increases exponentially at a rate of $(1 + \frac{1}{H})$. Specifically, let $e_1 = H$, and $e_{i+1} = \lfloor (1 + \frac{1}{H})e_i \rfloor, i \geq 1$ denote the lengths of the stages. Further, let the partial sums $\mathcal{L} \stackrel{\text{def}}{=} \{ \sum_{i=1}^j e_i \mid j = 1, 2, 3, \dots \}$ denote the set of the ending times of the stages. We remark that the stages are defined for each individual triple $(s, a, h)$, and for different triples the starting and ending times of their stages do not necessarily align in time.

Recall that the time index $(k, h)$ represents the $h$-th step of the $k$-th episode. At each step $(k, h)$, we take the optimal action with respect to the optimistic $Q_h(s, a)$ value (Line 6 in Algorithm 1), which is designed as an optimistic estimate of the optimal $Q_h^{k,\star}(s, a)$ value of the corresponding episode. For each triple $(s, a, h)$, we update the optimistic $Q_h(s, a)$ value at the end of each stage, using samples only from this latest stage that is about to end (Line 12 in Algorithm 1). The optimism in $Q_h(s, a)$ comes from two bonus terms $b_h^k$ and $b_\Delta$, where $b_h^k$ is a standard Hoeffding-based optimism that is commonly used in upper confidence bounds (Jin et al., 2018; Zhang et al., 2020), and $b_\Delta$ is the extra optimism (Cheung et al., 2020) that we need to take into account the non-stationarity of the environment. The definition of $b_\Delta$ requires knowledge of the local variation budget in each epoch, or at least an upper bound of it. The same assumption has also been made in Ortner et al. (2019). Fortunately, in our method, we can show (in Theorem 2) that if we simply replace Equation $(*)$ in Algorithm 1 with the following update rule:

$$Q_h(s_h^k, a_h^k) \leftarrow \min\left\{ \frac{\check{r}_h(s_h^k, a_h^k)}{\check{N}_h(s_h^k, a_h^k)} + \frac{\check{v}_h(s_h^k, a_h^k)}{\check{N}_h(s_h^k, a_h^k)} + b_h^k, Q_h(s_h^k, a_h^k) \right\} \tag{1}$$

then we can achieve the same regret bound without the assumption on the local variation budget. We set $\iota \stackrel{\text{def}}{=} \log\left(\frac{2}{\delta}\right)$, where $\delta$ is the failure probability.

## 4   ANALYSIS

In this section, we present our main result—a dynamic regret analysis of the RestartQ-UCB algorithm. Our first result on RestartQ-UCB with Hoeffding-style bonus terms is summarized in the following theorem. The complete proofs of its supporting lemmas are given in Appendix B.

**Theorem 1.** *(Hoeffding) For* $T = \Omega(SA\Delta H^2)$, *and for any* $\delta \in (0, 1)$, *with probability at least* $1 - \delta$, *the dynamic regret of RestartQ-UCB with Hoeffding bonuses (Algorithm 1) is bounded by* $\widetilde{O}(S^{\frac{1}{3}} A^{\frac{1}{3}} \Delta^{\frac{1}{3}} H^{\frac{5}{3}} T^{\frac{2}{3}})$, *where* $\widetilde{O}(\cdot)$ *hides poly-logarithmic factors of* $T$ *and* $1/\delta$.

Recall that we focus our analysis on epoch 1, with episode indices ranging from 1 to $K$. We start with the following technical lemma, stating that for any triple $(s, a, h)$, the difference of their optimal $Q$-values at two different episodes $1 \leq k_1 < k_2 \leq K$ is bounded by the variation of this epoch.

**Lemma 1.** *For any triple $(s, a, h)$ and any $1 \leq k_1 < k_2 \leq K$, it holds that $|Q_h^{k_1,\star}(s, a) - Q_h^{k_2,\star}(s, a)| \leq \Delta_r^{(1)} + H\Delta_p^{(1)}$.*

We now define a few notations to facilitate the analysis. Denote by $s_h^k$ and $a_h^k$ respectively the state and action taken at step $h$ of episode $k$. Let $N_h^k(s, a), \check{N}_h^k(s, a), Q_h^k(s, a)$ and $V_h^k(s)$ denote, respectively, the values of $N_h(s, a), \check{N}_h(s, a), Q_h(s, a)$ and $V_h(s)$ at the *beginning* of the $k$-th episode in Algorithm 1. Further, for the triple $(s_h^k, a_h^k, h)$, let $n_h^k$ be the total number of episodes that this triple has been visited prior to the current stage, and let $l_{h,i}^k$ denote the index of the episode that this triple was visited the $i$-th time among the total $n_h^k$ times. Similarly, let $\check{n}_h^k$ denote the number of visits to the triple $(s_h^k, a_h^k, h)$ in the stage right before the current stage, and let $\check{l}_{h,i}^k$ be the $i$-th episode among the $\check{n}_h^k$ episodes right before the current stage. For simplicity, we use $l_i$ and $\check{l}_i$ to denote $l_{h,i}^k$ and $\check{l}_{h,i}^k$, and $\check{n}$ to denote $\check{n}_h^k$, when $h$ and $k$ are clear from the context. We also use $\check{r}_h(s, a)$ and $\check{v}_h(s, a)$ to denote the values of $\check{r}_h(s_h^k, a_h^k)$ and $\check{v}_h(s_h^k, a_h^k)$ when updating the $Q_h(s_h^k, a_h^k)$ value in Line 12 of Algorithm 1.

The following lemma states that the optimistic $Q$-value $Q_h^k(s, a)$ is an upper bound of the optimal $Q$-value $Q_h^{k,\star}(s, a)$ with high probability. Note that we only need to show that the event holds with probability $1 - \texttt{poly}(K, H)\delta$, because we can replace $\delta$ with $\delta/\texttt{poly}(K, H)$ in the end to get the desired high probability bound without affecting the polynomial part of the regret bound.

**Lemma 2.** *(Hoeffding) For $\delta \in (0, 1)$, with probability at least $1 - 2KH\delta$, it holds that $Q_h^{k,\star}(s, a) \leq Q_h^{k+1}(s, a) \leq Q_h^k(s, a), \forall (s, a, h, k) \in \mathcal{S} \times \mathcal{A} \times [H] \times [K]$.*

We now proceed to analyze the dynamic regret in one epoch, and at the very end of this section, we will see how to combine the dynamic regret over all the epochs to prove Theorem 1. The following analysis will be conditioned on the successful event of Lemma 2.

The dynamic regret of Algorithm 1 in epoch $d = 1$ can hence be expressed as

$$\mathcal{R}^{(d)}(\pi, K) = \sum_{k=1}^K \left( V_1^{k,*}\left(s_1^k\right) - V_1^{k,\pi}\left(s_1^k\right) \right) \leq \sum_{k=1}^K \left( V_1^k\left(s_1^k\right) - V_1^{k,\pi}\left(s_1^k\right) \right). \tag{2}$$

From the update rules of the value functions in Algorithm 1, we have

$$V_h^k(s_h^k) \leq \mathbb{1}\left[n_h^k = 0\right] H + \frac{\check{r}_h(s_h^k, a_h^k)}{\check{N}_h^k(s_h^k, a_h^k)} + \frac{\check{v}_h(s_h^k, a_h^k)}{\check{N}_h^k(s_h^k, a_h^k)} + b_h^k + 2b_\Delta$$

$$= \mathbb{1}\left[n_h^k = 0\right] H + \frac{\check{r}_h(s_h^k, a_h^k)}{\check{N}_h^k(s_h^k, a_h^k)} + \frac{1}{\check{n}} \sum_{i=1}^{\check{n}} V_{h+1}^{\check{l}_i}(s_{h+1}^{\check{l}_i}) + b_h^k + 2b_\Delta.$$

For ease of exposition, we define the following notations:

$$\delta_h^k \overset{\text{def}}{=} V_h^k(s_h^k) - V_h^{k,\star}(s_h^k), \quad \zeta_h^k \overset{\text{def}}{=} V_h^k(s_h^k) - V_h^{k,\pi}(s_h^k). \tag{3}$$

We further define $\tilde{r}_h^k(s_h^k, a_h^k) \overset{\text{def}}{=} \frac{\check{r}_h(s_h^k, a_h^k)}{\check{N}_h^k(s_h^k, a_h^k)} - r_h^k(s_h^k, a_h^k)$. Then by the Hoeffding's inequality, it holds with high probability that

$$\tilde{r}_h^k(s_h^k, a_h^k) \leq \frac{1}{\check{n}} \sum_{i=1}^{\check{n}} r_h^{\check{l}_i}(s_h^k, a_h^k) + \sqrt{\frac{\iota}{\check{n}}} - r_h^k(s_h^k, a_h^k) \leq b_h^k + b_\Delta. \tag{4}$$

By the Bellman equation $V_h^{k,\pi}(s_h^k) = Q_h^{k,\pi}(s_h^k, \pi(s_h^k)) = r_h^k(s_h^k, a_h^k) + P_h^k V_{h+1}^{k,\pi}(s_h^k, a_h^k)$, we have

$$\zeta_h^k \leq \mathbb{1}\left[n_h^k = 0\right]H + \frac{1}{\check{n}}\sum_{i=1}^{\check{n}}V_{h+1}^{\check{l}_i}(s_{h+1}^{\check{l}_i}) + b_h^k + 2b_\Delta + \tilde{r}_h^k(s_h^k, a_h^k) - P_h^k V_{h+1}^{k,\pi}(s_h^k, a_h^k)$$

$$\leq \mathbb{1}\left[n_h^k = 0\right]H + \frac{1}{\check{n}}\sum_{i=1}^{\check{n}}P_h^{\check{l}_i}V_{h+1}^{\check{l}_i}(s_h^k, a_h^k) - P_h^k V_{h+1}^{k,\pi}(s_h^k, a_h^k) + 3b_h^k + 3b_\Delta \tag{5}$$

$$= \mathbb{1}\left[n_h^k = 0\right]H + \underbrace{\frac{1}{\check{n}}\sum_{i=1}^{\check{n}}\left(P_h^{\check{l}_i} - P_h^k\right)V_{h+1}^{\check{l}_i}(s_h^k, a_h^k)}_{①} + \underbrace{\frac{1}{\check{n}}\sum_{i=1}^{\check{n}}P_h^k\left(V_{h+1}^{\check{l}_i} - V_{h+1}^{\check{l}_i,\star}\right)(s_h^k, a_h^k)}_{②}$$

$$+ \underbrace{\frac{1}{\check{n}}\sum_{i=1}^{\check{n}}P_h^k\left(V_{h+1}^{\check{l}_i,\star} - V_{h+1}^{k,\pi}\right)(s_h^k, a_h^k)}_{③} + 3b_h^k + 3b_\Delta, \tag{6}$$

where (5) is by the Azuma-Hoeffding inequality and by (4). In the following, we bound each term in (6) separately. First, by Hölder's inequality, we have

$$① \leq \frac{1}{\check{n}}\sum_{i=1}^{\check{n}}\Delta_p^{(1)}(H - h) \leq b_\Delta. \tag{7}$$

Let $\mathbf{e}_j$ denote a standard basis vector of proper dimensions that has a 1 at the $j$-th entry and 0s at the others, in the form of $(0, \ldots, 0, 1, 0, \ldots, 0)$. Recall the definition of $\delta_h^k$ in (3), and we have

$$② = \frac{1}{\check{n}}\sum_{i=1}^{\check{n}}\delta_{h+1}^{\check{l}_i} + \underbrace{\frac{1}{\check{n}}\sum_{i=1}^{\check{n}}\left(P_h^k - \mathbf{e}_{s_{h+1}^{\check{l}_i}}\right)\left(V_{h+1}^{\check{l}_i} - V_{h+1}^{\check{l}_i,\star}\right)(s_h^k, a_h^k)}_{\xi_{h+1}^k} = \frac{1}{\check{n}}\sum_{i=1}^{\check{n}}\delta_{h+1}^{\check{l}_i} + \xi_{h+1}^k. \tag{8}$$

Finally, recalling the definition of $\zeta_h^k$ in (3), we have that

$$③ = \frac{1}{\check{n}}\sum_{i=1}^{\check{n}}\left(V_{h+1}^{\check{l}_i,\star}(s_{h+1}^k) - V_{h+1}^{k,\pi}(s_{h+1}^k)\right) + \underbrace{\frac{1}{\check{n}}\sum_{i=1}^{\check{n}}\left(P_h^k - \mathbf{e}_{s_{h+1}^k}\right)\left(V_{h+1}^{\check{l}_i,\star} - V_{h+1}^{k,\pi}\right)(s_h^k, a_h^k)}_{\phi_{h+1}^k}$$

$$= \frac{1}{\check{n}}\sum_{i=1}^{\check{n}}\left(V_{h+1}^{\check{l}_i,\star}(s_{h+1}^k) - V_{h+1}^{k,\star}(s_{h+1}^k)\right) + \zeta_{h+1}^k - \delta_{h+1}^k + \phi_{h+1}^k$$

$$\leq b_\Delta + \zeta_{h+1}^k - \delta_{h+1}^k + \phi_{h+1}^k \tag{9}$$

where inequality (9) is by Lemma 1. Combining (6), (7), (8), and (9) leads to

$$\zeta_h^k \leq \mathbb{1}\left[n_h^k = 0\right]H + \frac{1}{\check{n}}\sum_{i=1}^{\check{n}}\delta_{h+1}^{\check{l}_i} + \xi_{h+1}^k + \zeta_{h+1}^k - \delta_{h+1}^k + \phi_{h+1}^k + 3b_h^k + 5b_\Delta. \tag{10}$$

To find an upper bound of $\sum_{k=1}^K \zeta_h^k$, we proceed to upper bound each term on the RHS of (10) separately. First, notice that $\sum_{k=1}^K \mathbb{1}\left[n_h^k = 0\right] \leq SAH$, because each fixed triple $(s, a, h)$ contributes at most 1 to $\sum_{k=1}^K \mathbb{1}\left[n_h^k = 0\right]$. The second term in (10) can be upper bounded by the following lemma:

**Lemma 3.** $\sum_{k=1}^K \frac{1}{\check{n}_h^k}\sum_{i=1}^{\check{n}_h^k}\delta_{h+1}^{\check{l}_{h,i}^k} \leq (1 + \frac{1}{H})\sum_{k=1}^K \delta_{h+1}^k$.

Combining (10) and Lemma 3, we now have that

$$\sum_{k=1}^{K} \zeta_h^k \leq SAH^2 + \frac{1}{H} \sum_{k=1}^{K} \delta_{h+1}^k + \sum_{k=1}^{K} \left( \xi_{h+1}^k + \zeta_{h+1}^k + \phi_{h+1}^k + 3b_h^k + 5b_\Delta \right)$$

$$\leq SAH^2 + (1 + \frac{1}{H}) \sum_{k=1}^{K} \zeta_{h+1}^k + \sum_{k=1}^{K} \underbrace{\left( \xi_{h+1}^k + \phi_{h+1}^k + 3b_h^k + 5b_\Delta \right)}_{\Lambda_{h+1}^k}, \qquad (11)$$

where in (11) we have used the fact that $\delta_{h+1}^k \leq \zeta_{h+1}^k$, which in turn is due to the optimality that $V_h^{k,\star}(s_h^k) \geq V_h^{k,\pi}(s_h^k)$. Notice that we have $\zeta_h^k$ on the LHS of (11) and $\zeta_{h+1}^k$ on the RHS. By iterating (11) over $h = H, H-1, \ldots, 1$, we conclude that

$$\sum_{k=1}^{K} \zeta_1^k \leq O \left( SAH^3 + \sum_{h=1}^{H} \sum_{k=1}^{K} (1 + \frac{1}{H})^{h-1} \Lambda_{h+1}^k \right). \qquad (12)$$

We bound $\sum_{h=1}^{H} \sum_{k=1}^{K} (1 + \frac{1}{H})^{h-1} \Lambda_{h+1}^k$ in the proposition below. Its proof relies on a series of lemmas in Appendix B that upper bound each term in $\Lambda_{h+1}^k$ separately.

**Proposition 1.** *With probability at least* $1 - (KH + 2)\delta$, *it holds that*

$$\sum_{h=1}^{H} \sum_{k=1}^{K} (1 + \frac{1}{H})^{h-1} \Lambda_{h+1}^k \leq \widetilde{O} \left( \sqrt{SAKH^5} + KH\Delta_r^{(1)} + KH^2\Delta_p^{(1)} \right).$$

Now we are ready to prove Theorem 1.

*Proof.* (of Theorem 1) By (2) and (12), and by replacing $\delta$ with $\frac{\delta}{KH+2}$ in Proposition 1, we know that the dynamic regret in epoch $d = 1$ can be upper bounded with probability at least $1 - \delta$ by:

$$\mathcal{R}^{(d)}(\pi, K) \leq \widetilde{O} \left( SAH^3 + \sqrt{SAKH^5} + KH\Delta_r^{(1)} + KH^2\Delta_p^{(1)} \right),$$

and this holds for every epoch $d \in [D]$. Suppose $T = \Omega(SA\Delta H^2)$; summing up the dynamic regret over all the $D$ epochs gives us an upper bound of $\widetilde{O}(D\sqrt{SAKH^5} + \sum_{d=1}^{D} KH\Delta_r^{(d)} + \sum_{d=1}^{D} KH^2\Delta_p^{(d)})$. Recall the definition that $\sum_{d=1}^{D} \Delta_r^{(d)} \leq \Delta_r$, $\sum_{d=1}^{D} \Delta_p^{(d)} \leq \Delta_p$, $\Delta = \Delta_r + \Delta_p$, and that $K = \Theta(\frac{T}{DH})$. By setting $D = S^{-\frac{1}{3}} A^{-\frac{1}{3}} \Delta^{\frac{2}{3}} H^{-\frac{2}{3}} T^{\frac{1}{3}}$, the dynamic regret over the entire $T$ steps is bounded by $\mathcal{R}(\pi, M) \leq \widetilde{O}(S^{\frac{1}{3}} A^{\frac{1}{3}} \Delta^{\frac{1}{3}} H^{\frac{5}{3}} T^{\frac{2}{3}})$, which completes the proof. $\qquad \square$

Algorithm 1 relies on the assumption that the local budgets $b_\Delta$ are known a priori, which hardly holds in practice. In the following theorem, we will show that this assumption can be safely removed without affecting the regret bound. The only modification to the algorithm is to replace the $Q$-value update rule in Equation $(*)$ of Algorithm 1 with the new update rule in Equation (1).

**Theorem 2.** *(Hoeffding, no local budgets) For* $T = \Omega(SA\Delta H^2)$, *and for any* $\delta \in (0, 1)$, *with probability at least* $1 - \delta$, *the dynamic regret of RestartQ-UCB with Hoeffding bonuses and no knowledge of local budgets is bounded by* $\widetilde{O}(S^{\frac{1}{3}} A^{\frac{1}{3}} \Delta^{\frac{1}{3}} H^{\frac{5}{3}} T^{\frac{2}{3}})$, *where* $\widetilde{O}(\cdot)$ *hides poly-logarithmic factors of* $T$ *and* $1/\delta$.

To understand why this simple modification works, notice that in $(*)$ we are adding exactly the same value $2b_\Delta$ to the upper confidence bounds of all $(s, a)$ pairs in the same epoch. Subtracting the same value from all optimistic $Q$-values simultaneously should not change the choice of actions in future steps. The only difference is that the new "optimistic" $Q_h^k(s, a)$ values would no longer be strict upper bounds of the optimal $Q_h^{k,\star}(s, a)$ anymore, but only an "upper bound" subject to some error term of the order $b_\Delta$. This further requires a slightly different analysis on how this error term propagates over time, which is presented as a variant of Lemma 2 as follows.

**Lemma 4.** *(Hoeffding, no local budgets) Suppose we have no knowledge of the local variation budgets and replace the update rule* (∗) *in Algorithm 1 with Equation* (1)*. For $\delta \in (0, 1)$, with probability at least $1 - 2KH\delta$, it holds that $Q_h^{k,\star}(s, a) - 2(H - h + 1)b_\Delta \leq Q_h^{k+1}(s, a) \leq Q_h^k(s, a), \forall(s, a, h, k) \in \mathcal{S} \times \mathcal{A} \times [H] \times [K]$.*

*Remark* 1. The easy removal of the local budget assumption is non-trivial in the design of the algorithm, and does not exist in the non-stationary RL literature with restarts. In fact, it has been shown in a concurrent work (Zhou et al., 2020) that removing this assumption would lead to a much worse regret bound (cf. Corollary 2 and Corollary 3 therein).

Replacing the Hoeffding-based upper confidence bound with a Freedman-style one will lead to a tighter regret bound, summarized in Theorem 3 below. The proof of the theorem follows a similar procedure as in the proof of Theorem 1, and is given in Appendix D. It relies on a reference-advantage decomposition technique for variance reduction as coined in Zhang et al. (2020). The intuition is to first learn a reference value function $V^{\text{ref}}$ that serves as a roughly accurate estimate of the optimal value function $V^\star$. The goal of learning the optimal value function $V^\star = V^{\text{ref}} + (V^* - V_{\text{ref}})$ can hence be decomposed into estimating two terms $V^{\text{ref}}$ and $V^* - V_{\text{ref}}$, each of which can be accurately estimated due to the reduced variance. For ease of exposition, we proceed again with the assumption that the local variation budgets are known. The reader should bear in mind that this assumption can be easily removed using a similar technique as in Theorem 2.

**Theorem 3.** *(Freedman) For $T$ greater than some polynomial of $S, A, \Delta$ and $H$, and for any $\delta \in (0, 1)$, with probability at least $1 - \delta$, the dynamic regret of RestartQ-UCB with Freedman bonuses (Algorithm 2) is bounded by $\widetilde{O}(S^{\frac{1}{3}}A^{\frac{1}{3}}\Delta^{\frac{1}{3}}HT^{\frac{2}{3}})$, where $\widetilde{O}(\cdot)$ hides poly-logarithmic factors.*

## 5 LOWER BOUNDS

In this section, we provide information-theoretical lower bounds of the dynamic regret to characterize the best achievable performance of any algorithm for solving non-stationary MDPs.

**Theorem 4.** *For any algorithm, there exists an episodic non-stationary MDP such that the dynamic regret of the algorithm is at least $\Omega(S^{\frac{1}{3}}A^{\frac{1}{3}}\Delta^{\frac{1}{3}}H^{\frac{2}{3}}T^{\frac{2}{3}})$.*

*Proof sketch.* The proof of our lower bound relies on the construction of a "hard instance" of non-stationary MDPs. The instance we construct is essentially a switching-MDP: an MDP with piece-wise constant dynamics on each *segment* of the horizon, and its dynamics experience an abrupt change at the beginning of each new segment. More specifically, we divide the horizon $T$ into $L$ segments[1], where each segment has $T_0 \overset{\text{def}}{=} \lfloor \frac{T}{L} \rfloor$ steps and contains $M_0 \overset{\text{def}}{=} \lfloor \frac{M}{L} \rfloor$ episodes, each episode having a length of $H$. Within each such segment, the system dynamics of the MDP do not vary, and we construct the dynamics for each segment in a way such that the instance is a hard instance of stationary MDPs on its own. The MDP within each segment is essentially similar to the hard instances constructed in stationary RL problems (Osband & Van Roy, 2016; Jin et al., 2018). Between two consecutive segments, the dynamics of the MDP change abruptly, and we let the dynamics vary in a way such that no information learned from previous interactions with the MDP can be used in the new segment. In this sense, the agent needs to learn a new hard stationary MDP in each segment. Finally, optimizing the value of $L$ and the variation magnitude between consecutive segments (subject to the constraints of the total variation budget) leads to our lower bound. □

A useful side result of our proof is the following lower bound for non-stationary RL in the un-discounted setting, which is the same setting as studied in Cheung et al. (2020), Gajane et al. (2018) and Ortner et al. (2019).

**Proposition 2.** *Consider a reinforcement learning problem in un-discounted non-stationary MDPs with horizon length $T$, total variation budget $\Delta$, and maximum MDP diameter $D$ (Cheung et al., 2020). For any learning algorithm, there exists a non-stationary MDP such that the dynamic regret of the algorithm is at least $\Omega(S^{\frac{1}{3}}A^{\frac{1}{3}}\Delta^{\frac{1}{3}}D^{\frac{2}{3}}T^{\frac{2}{3}})$.*

---

[1]The definition of segments is irrelevant to, and should not be confused with, the notion of epochs we previously defined.

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

# A    APPLICATIONS TO SEQUENTIAL TRANSFER, MULTI-TASK, AND MULTI-AGENT RL

One area that could benefit from non-stationary RL is sequential transfer in RL (Tirinzoni et al., 2020) or multi-task RL (Brunskill & Li, 2013), which itself is conceptually related to continual RL (Kaplanis et al., 2018) and life-long RL (Abel et al., 2018). In the setting of sequential transfer/multi-task RL, the agent encounters a sequence of tasks over time with different system dynamics, and seeks to bootstrap learning by transferring knowledge from previously-solved tasks. Typical solutions in this area (Brunskill & Li, 2013; Tirinzoni et al., 2020; Sun et al., 2020) need to assume that there are *finitely many* candidate tasks, and every task should be *sufficiently different* from the others[2]. Only under this assumption can the agent quickly identify the current task it is operating on, by essentially comparing the system dynamics it observes with the dynamics it has memorized for each candidate task. After identifying the current task with high confidence, the agent then invokes the policy that it learned through previous interactions with this specific task. This transfer learning paradigm in turn causes another problem—it "cold switches" between policies that are most likely very different, which might lead to unstable and inconsistent behaviors of the agent over time. Fortunately, non-stationary RL can help alleviate both the finite-task assumption and the cold-switching problem. First, non-stationary RL algorithms do not need the candidate tasks to be sufficiently different in order to correctly identify each of them, because the algorithm itself can tolerate some variations in the task environment. There will also be no need to assume the finiteness of the candidate task set anymore, and the candidate tasks can be drawn from a continuous space. Second, since we are running the same non-stationary RL algorithm for a series of tasks, it improves its policy gradually over time, instead of cold-switching to a completely independent policy for each task. This could largely help with the unstable behavior issues.

Multi-agent reinforcement learning (MARL) (Littman, 1994) studies the problem where a set of agents collaborate or compete in a shared environment. In MARL, since the transition and reward functions of the agents are coupled, the environment is non-stationary from each agent's own perspective, especially when the agents learn and update their policies simultaneously. The non-stationarity in MARL is a setting where non-stationary RL can play a role. As advocated earlier in Bowling & Veloso (2001); Busoniu et al. (2008), a good MARL algorithm should be both *rational* and *convergent*, where the former means that the algorithm converges to its opponent's *best response* if its opponent converges to a stationary policy, and the latter means that if all agents use the same algorithm, the algorithm converges to a stationary policy. As such, a non-stationary RL algorithm can be viewed as a *rational* MARL algorithm, thanks to its dynamic regret guarantees, although its *convergence* property in MARL settings is still worth further investigation. In fact, developing algorithms that are both *rational* and *convergent* in general MARL settings is still relatively open. In addition, non-stationary RL algorithms also apply to the MARL setting to achieve low regret against *slowly-changing* opponents (see (Lee et al., 2020, Sec. 5.2) for the setting) but we consider a more challenging measure of dynamic regret (as opposed to the static regret in Lee et al. (2020)). Finally, dynamic regret is also pertinent to the notion of *exploitability* of strategies in two-player zero-sum games (Davis et al., 2014).

# B    PROOFS OF THE TECHNICAL LEMMAS

## B.1    PROOF OF LEMMA 1

*Proof.* In fact, in the following, we will prove a stronger statement: $\left| Q_h^{k_1,\star}(s,a) - Q_h^{k_2,\star}(s,a) \right| \le \sum_{h'=h}^{H} \Delta_{r,h'}^{(1)} + H \sum_{h'=h}^{H} \Delta_{p,h'}^{(1)}$, which implies the statement of the lemma because $\sum_{h'=h}^{H} \Delta_{r,h'}^{(1)} \le \Delta_r^{(1)}$ and $\sum_{h'=h}^{H} \Delta_{p,h'}^{(1)} \le \Delta_p^{(1)}$ by definition. Our proof relies on backward induction on $h$. First,

---

[2]Needless to say, this assumption itself also to some extent contradicts the primary motivation of transfer learning. After all, we only want to transfer knowledge among tasks that are essentially similar to each other.

the statement holds for $h = H$ because for any $(s, a)$, by definition

$$\left| Q_H^{k_1,\star}(s,a) - Q_H^{k_2,\star}(s,a) \right| = \left| r_H^{k_1}(s,a) - r_H^{k_2}(s,a) \right| \leq \sum_{k=k_1}^{k_2-1} \left| r_H^{k+1}(s,a) - r_H^k(s,a) \right|$$

$$\leq \sum_{k=1}^{K-1} \left| r_H^{k+1}(s,a) - r_H^k(s,a) \right| \leq \Delta_{r,H}^{(1)}, \tag{13}$$

where we have used the triangle inequality. Now suppose the statement holds for $h + 1$; by the Bellman optimality equation,

$$Q_h^{k_1,\star}(s,a) - Q_h^{k_2,\star}(s,a)$$
$$= P_h^{k_1} V_{h+1}^{k_1,\star}(s,a) - P_h^{k_2} V_{h+1}^{k_2,\star}(s,a) + r_h^{k_1}(s,a) - r_h^{k_2}(s,a)$$
$$\leq P_h^{k_1} V_{h+1}^{k_1,\star}(s,a) - P_h^{k_1} V_{h+1}^{k_2,\star}(s,a) + \Delta_{r,h}^{(1)} \tag{14}$$
$$= \sum_{s' \in \mathcal{S}} P_h^{k_1}(s' \mid s,a) V_{h+1}^{k_1,\star}(s') - \sum_{s' \in \mathcal{S}} P_h^{k_2}(s' \mid s,a) V_{h+1}^{k_2,\star}(s') + \Delta_{r,h}^{(1)}$$
$$= \sum_{s' \in \mathcal{S}} \left( P_h^{k_1}(s' \mid s,a) Q_{h+1}^{k_1,\star}(s', \pi_{h+1}^{k_1,\star}(s')) - P_h^{k_2}(s' \mid s,a) Q_{h+1}^{k_2,\star}(s', \pi_{h+1}^{k_2,\star}(s')) \right) + \Delta_{r,h}^{(1)}, \tag{15}$$

where inequality (14) holds due to a similar reasoning as in (13), and in (15) $\pi^{k_1,\star}$ and $\pi^{k_2,\star}$ denote the optimal policy in episode $k_1$ and $k_2$, respectively. Then by our induction hypothesis on $h + 1$, for any $s' \in \mathcal{S}$,

$$Q_{h+1}^{k_1,\star}(s', \pi_{h+1}^{k_1,\star}(s')) \leq Q_{h+1}^{k_2,\star}(s', \pi_{h+1}^{k_1,\star}(s')) + \sum_{h'=h+1}^{H} \Delta_{r,h'}^{(1)} + H \sum_{h'=h+1}^{H} \Delta_{p,h'}^{(1)}$$

$$\leq Q_{h+1}^{k_2,\star}(s', \pi_{h+1}^{k_2,\star}(s')) + \sum_{h'=h+1}^{H} \Delta_{r,h'}^{(1)} + H \sum_{h'=h+1}^{H} \Delta_{p,h'}^{(1)}, \tag{16}$$

where inequality (16) is due to the optimality of the policy $\pi^{k_2,\star}$ in episode $k_2$ over $\pi^{k_1,\star}$. Then,

$$Q_h^{k_1,\star}(s,a) - Q_h^{k_2,\star}(s,a)$$

$$\leq \sum_{s' \in \mathcal{S}} (P_h^{k_1}(s' \mid s,a) - P_h^{k_2}(s' \mid s,a)) Q_{h+1}^{k_2,\star}(s', \pi_{h+1}^{k_2,\star}(s')) + \sum_{h'=h}^{H} \Delta_{r,h'}^{(1)} + H \sum_{h'=h+1}^{H} \Delta_{p,h'}^{(1)}$$

$$\leq \left\| P_h^{k_1}(\cdot | s,a) - P_h^{k_2}(\cdot | s,a) \right\|_1 \left\| Q_{h+1}^{k_2,\star}(\cdot, \pi_{h+1}^{k_2,\star}(\cdot)) \right\|_\infty + \sum_{h'=h}^{H} \Delta_{r,h'}^{(1)} + H \sum_{h'=h+1}^{H} \Delta_{p,h'}^{(1)} \tag{17}$$

$$\leq \Delta_{p,h}^{(1)}(H-h) + \sum_{h'=h}^{H} \Delta_{r,h'}^{(1)} + H \sum_{h'=h+1}^{H} \Delta_{p,h'}^{(1)} \tag{18}$$

$$\leq \sum_{h'=h}^{H} \Delta_{r,h'}^{(1)} + H \sum_{h'=h}^{H} \Delta_{p,h'}^{(1)},$$

where (17) is by Hölder's inequality, and (18) is by the definition of $\Delta_{p,h}^{(1)}$ and by the definition of optimal $Q$-values that $Q_{h+1}^{k_2,\star}(s,a) \leq H - h, \forall (s,a) \in \mathcal{S} \times \mathcal{A}$. Repeating a similar process gives us $Q_h^{k_2,\star}(s,a) - Q_h^{k_1,\star}(s,a) \leq \sum_{h'=h}^{H} \Delta_{r,h'}^{(1)} + H \sum_{h'=h}^{H} \Delta_{p,h'}^{(1)}$. This completes our proof. $\qquad \square$

### B.2  PROOF OF LEMMA 2

*Proof.* It should be clear from the way we update $Q_h(s,a)$ that $Q_h^k(s,a)$ is monotonically decreasing in $k$. We now prove $Q_h^{k,\star}(s,a) \leq Q_h^{k+1}(s,a)$ for all $s, a, h, k$ by induction on $k$. First, it holds for $k = 1$ by our initialization of $Q_h(s,a)$. For $k \geq 2$, now suppose $Q_h^{j,\star}(s,a) \leq Q_h^{j+1}(s,a) \leq$

$Q_h^j(s, a)$ for all $s, a, h$ and $1 \leq j \leq k$. For a fixed triple $(s, a, h)$, we consider the following two cases.

**Case 1:** $Q_h(s, a)$ is updated in episode $k$. Then with probability at least $1 - 2\delta$

$$Q_h^{k+1}(s, a) = \frac{\check{r}_h(s, a)}{\check{N}_h^k(s, a)} + \frac{\check{v}_h(s, a)}{\check{N}_h^k(s, a)} + b_h^k + 2b_\Delta$$

$$\geq \frac{\check{r}_h(s, a)}{\check{n}} + \frac{1}{\check{n}} \sum_{i=1}^{\check{n}} V_{h+1}^{\check{l}_i, \star}(s_{h+1}^{\check{l}_i}) + \sqrt{\frac{H^2}{\check{n}} \iota} + \sqrt{\frac{\iota}{\check{n}}} + 2b_\Delta \qquad (19)$$

$$\geq \frac{\check{r}_h(s, a)}{\check{n}} + \frac{1}{\check{n}} \sum_{i=1}^{\check{n}} P_h^{\check{l}_i} V_{h+1}^{\check{l}_i, \star}(s, a) + \sqrt{\frac{\iota}{\check{n}}} + 2b_\Delta \qquad (20)$$

$$= \frac{\check{r}_h(s, a)}{\check{n}} + \frac{1}{\check{n}} \sum_{i=1}^{\check{n}} \left( Q_h^{\check{l}_i, \star}(s, a) - r_h^{\check{l}_i}(s, a) \right) + \sqrt{\frac{\iota}{\check{n}}} + 2b_\Delta \qquad (21)$$

$$\geq Q_h^{k, \star}(s, a) + b_\Delta. \qquad (22)$$

Inequality (19) is by the induction hypothesis that $Q_{h+1}^{\check{l}_i}(s_{h+1}^{\check{l}_i}, a) \geq Q_{h+1}^{\check{l}_i, \star}(s_{h+1}^{\check{l}_i}, a), \forall a \in \mathcal{A}$, and hence $V_{h+1}^{\check{l}_i}(s_{h+1}^{\check{l}_i}) \geq V_{h+1}^{\check{l}_i, \star}(s_{h+1}^{\check{l}_i})$. Inequality (20) follows from the Azuma-Hoeffding inequality. (21) uses the Bellman optimality equation. Inequality (22) is by the Hoeffding's inequality that $\frac{1}{\check{n}} \left( \sum_{i=1}^{\check{n}} r_h^{\check{l}_i}(s, a) - \check{r}_h(s, a) \right) \leq \sqrt{\frac{\iota}{\check{n}}}$ with high probability, and by Lemma 1 that $Q_h^{\check{l}_i, \star}(s, a) + b_\Delta \geq Q_h^{k, \star}(s, a)$. According to the monotonicity of $Q_h^k(s, a)$, we know that $Q_h^{k, \star}(s, a) \leq Q_h^{k+1}(s, a) \leq Q_h^k(s, a)$. In fact, we have proved the stronger statement $Q_h^{k+1}(s, a) \geq Q_h^{k, \star}(s, a) + b_\Delta$ that will be useful in Case 2 below.

**Case 2:** $Q_h(s, a)$ is not updated in episode $k$. Then there are two possibilities:

1. If $Q_h(s, a)$ has never been updated from episode 1 to episode $k$: It is easy to see that $Q_h^{k+1}(s, a) = Q_h^k(s, a) = \cdots = Q_h^1(s, a) = H - h + 1 \geq Q_h^{k, \star}(s, a)$ holds.

2. If $Q_h(s, a)$ has been updated at least once from episode 1 to episode $k$: Let $j$ be the index of the latest episode that $Q_h(s, a)$ was updated. Then, from our induction hypothesis and Case 1, we know that $Q_h^{j+1}(s, a) \geq Q_h^{j, \star}(s, a) + b_\Delta$. Since $Q_h(s, a)$ has not been updated from episode $j + 1$ to episode $k$, we know that $Q_h^{k+1}(s, a) = Q_h^k(s, a) = \cdots = Q_h^{j+1}(s, a) \geq Q_h^{j, \star}(s, a) + b_\Delta \geq Q_h^{k, \star}(s, a)$, where the last inequality holds because of Lemma 1.

A union bound over all time steps completes our proof. □

## B.3 Proof of Lemma 3

*Proof.* It holds that

$$\sum_{k=1}^{K} \frac{1}{\check{n}_h^k} \sum_{i=1}^{\check{n}_h^k} \delta_{h+1}^{\check{l}_{h,i}^k} = \sum_{k=1}^{K} \sum_{j=1}^{K} \frac{1}{\check{n}_h^k} \delta_{h+1}^j \sum_{i=1}^{\check{n}_h^k} \mathbb{1}\left[ \check{l}_{h,i}^k = j \right] = \sum_{j=1}^{K} \delta_{h+1}^j \sum_{k=1}^{K} \frac{1}{\check{n}_h^k} \sum_{i=1}^{\check{n}_h^k} \mathbb{1}\left[ \check{l}_{h,i}^k = j \right]. \quad (23)$$

For a fixed episode $j$, notice that $\sum_{i=1}^{\check{n}_h^k} \mathbb{1}[\check{l}_{h,i}^k = j] \leq 1$, and that $\sum_{i=1}^{\check{n}_h^k} \mathbb{1}[\check{l}_{h,i}^k = j] = 1$ happens if and only if $(s_h^k, a_h^k) = (s_h^j, a_h^j)$ and $(j, h)$ lies in the previous stage of $(k, h)$ with respect to the triple $(s_h^k, a_h^k, h)$. Let $\mathcal{K} \stackrel{\text{def}}{=} \{k \in [K] : \sum_{i=1}^{\check{n}_h^k} \mathbb{1}[\check{l}_{h,i}^k = j] = 1\}$; then, we know that every element $k \in \mathcal{K}$ has the same value of $\check{n}_h^k$, i.e., there exists an integer $N_j > 0$, such that $\check{n}_h^k = N_j, \forall k \in \mathcal{K}$. Further, by our definition of the stages, we know that $|\mathcal{K}| \leq (1 + \frac{1}{H}) N_j$, because the current stage

is at most $(1 + \frac{1}{H})$ times longer than the previous stage. Therefore, for every $j$, we know that

$$\sum_{k=1}^{K} \frac{1}{\check{n}_h^k} \sum_{i=1}^{\check{n}_h^k} \mathbb{1} \left[ \check{l}_{h,i}^k = j \right] \leq 1 + \frac{1}{H}. \tag{24}$$

Combining (23) and (24) completes the proof of $\sum_{k=1}^{K} \frac{1}{\check{n}_h^k} \sum_{i=1}^{\check{n}_h^k} \delta_{h+1}^{\check{l}_{h,i}^k} \leq (1 + \frac{1}{H}) \sum_{k=1}^{K} \delta_{h+1}^k$. □

### B.4 PROOF OF PROPOSITION 1

In the following, we will bound each term in $\Lambda_{h+1}^k$ separately in a series of lemmas.

**Lemma 5.** *With probability* 1*, we have that*

$$\sum_{h=1}^{H} \sum_{k=1}^{K} (1 + \frac{1}{H})^{h-1} (3b_h^k + 5b_\Delta) \leq O(\sqrt{SAKH^5\iota} + KH\Delta_r^{(1)} + KH^2\Delta_p^{(1)}).$$

*Proof.* First, by the definition of $b_\Delta$, it is easy to see that $\sum_{h=1}^{H} \sum_{k=1}^{K} (1 + \frac{1}{H})^{h-1} 5b_\Delta \leq \sum_{h=1}^{H} \sum_{k=1}^{K} O(\Delta_r^{(1)} + H\Delta_p^{(1)}) \leq O(KH\Delta_r^{(1)} + KH^2\Delta_p^{(1)})$. Recall our definition that $e_1 = H$ and $e_{i+1} = \lfloor (1 + \frac{1}{H})e_i \rfloor, i \geq 1$. For a fixed $h \in [H]$, since $H^2 \geq 1$,

$$\sum_{k=1}^{K} (1 + \frac{1}{H})^{h-1} 3b_h^k \leq \sum_{k=1}^{K} (1 + \frac{1}{H})^{h-1} 12 \sqrt{\frac{H^2}{\check{N}_h^k(s_h^k, a_h^k)} \iota}$$

$$= 12H\sqrt{\iota} \sum_{s,a} \sum_{j \geq 1} (1 + \frac{1}{H})^{h-1} \sqrt{\frac{1}{e_j} \sum_{k=1}^{K} \mathbb{1} \left[ (s_h^k, a_h^k) = (s,a), \check{N}_h^k(s_h^k, a_h^k) = e_j \right]}$$

$$= 12H\sqrt{\iota} \sum_{s,a} \sum_{j \geq 1} (1 + \frac{1}{H})^{h-1} w(s,a,j) \sqrt{\frac{1}{e_j}},$$

where $w(s,a,j) \stackrel{\text{def}}{=} \sum_{k=1}^{K} \mathbb{1} \left[ (s_h^k, a_h^k) = (s,a), \check{N}_h^k(s_h^k, a_h^k) = e_j \right]$, and $w(s,a) \stackrel{\text{def}}{=} \sum_{j \geq 1} w(s,a,j)$. We then know that $\sum_{s,a} w(s,a) = K$. For a fixed $(s,a)$, let us now find an upper bound of $j$, denoted as $J$. Since each stage is $(1 + \frac{1}{H})$ times longer than the previous stage, we know for $1 \leq j \leq J$, $w(s,a,j) = \sum_{k=1}^{K} \mathbb{1} \left[ (s_h^k, a_h^k) = (s,a), \check{N}_h^k(s_h^k, a_h^k) = e_j \right] = \lfloor (1 + \frac{1}{H})e_j \rfloor$. From $\sum_{j=1}^{J} w(s,a,j) = w(s,a)$, we get $e_J \leq (1 + \frac{1}{H})^{J-1} \leq \frac{10}{1+\frac{1}{H}} \frac{w(s,a)}{H}$. Therefore,

$$\sum_{j \geq 1} (1 + \frac{1}{H})^{h-1} w(s,a,j) \sqrt{\frac{1}{e_j}} \leq O \left( \sum_{j=1}^{J} \sqrt{e_j} \right) \leq O \left( \sqrt{w(s,a)H} \right).$$

Finally, by the Cauchy-Schwartz inequality, we have

$$\sum_{h=1}^{H} \sum_{k=1}^{K} (1 + \frac{1}{H})^{h-1} 3b_h^k = O \left( H^2 \sqrt{\iota} \sum_{s,a} \sum_{j \geq 1} w(s,a,j) \sqrt{\frac{1}{e_j}} \right) \leq \sqrt{SAKH^5\iota}.$$

Combining the bounds for $b_h^k$ and $b_\Delta$ completes the proof. □

**Lemma 6.** *With probability at least* $1 - \delta$*, it holds that*

$$\sum_{h=1}^{H} \sum_{k=1}^{K} (1 + \frac{1}{H})^{h-1} \phi_{h+1}^k \leq O(\sqrt{KH^3\iota} + KH\Delta_r^{(1)} + KH^2\Delta_p^{(1)}).$$

*Proof.* We have that

$$\sum_{h=1}^{H}\sum_{k=1}^{K}(1+\frac{1}{H})^{h-1}\phi_{h+1}^{k}$$

$$=\sum_{h=1}^{H}\sum_{k=1}^{K}(1+\frac{1}{H})^{h-1}\frac{1}{\check{n}}\sum_{i=1}^{\check{n}}\left(P_{h}^{k}-\mathbf{e}_{s_{h+1}^{k}}\right)\left(V_{h+1}^{\check{l}_{i},\star}-V_{h+1}^{k,\pi}\right)(s_{h}^{k},a_{h}^{k})$$

$$=\sum_{h=1}^{H}\sum_{k=1}^{K}(1+\frac{1}{H})^{h-1}\frac{1}{\check{n}}\sum_{i=1}^{\check{n}}\left(P_{h}^{k}-\mathbf{e}_{s_{h+1}^{k}}\right)\left(V_{h+1}^{\check{l}_{i},\star}-V_{h+1}^{k,\star}+V_{h+1}^{k,\star}-V_{h+1}^{k,\pi}\right)(s_{h}^{k},a_{h}^{k})$$

$$\leq\sum_{h=1}^{H}\sum_{k=1}^{K}(1+\frac{1}{H})^{h-1}2b_{\Delta}+\sum_{h=1}^{H}\sum_{k=1}^{K}(1+\frac{1}{H})^{h-1}\left(P_{h}^{k}-\mathbf{e}_{s_{h+1}^{k}}\right)\left(V_{h+1}^{k,\star}-V_{h+1}^{k,\pi}\right)(s_{h}^{k},a_{h}^{k}),$$

where the last inequality follows from Lemma 1 and the definition of $b_{\Delta}$. From the proof of Lemma 5, we know that the first term can be bounded as

$$\sum_{h=1}^{H}\sum_{k=1}^{K}(1+\frac{1}{H})^{h-1}2b_{\Delta}\leq O(KH\Delta_{r}^{(1)}+KH^{2}\Delta_{p}^{(1)}).$$

Further, the second term is bounded by the Azuma-Hoeffding inequality as

$$\sum_{h=1}^{H}\sum_{k=1}^{K}(1+\frac{1}{H})^{h-1}\left(P_{h}^{k}-\mathbf{e}_{s_{h+1}^{k}}\right)\left(V_{h+1}^{k,\star}-V_{h+1}^{k,\pi}\right)(s_{h}^{k},a_{h}^{k})\leq O(\sqrt{KH^{3}\iota}).$$

Combining the two terms completes the proof. $\qquad\square$

**Lemma 7.** *With probability at least $1-(KH+1)\delta$, it holds that*

$$\sum_{h=1}^{H}\sum_{k=1}^{K}(1+\frac{1}{H})^{h-1}\xi_{h+1}^{k}\leq O(\sqrt{SAKH^{3}\iota}+KH^{2}\Delta_{p}^{(1)}).$$

*Proof.* We have that

$$\sum_{h=1}^{H}\sum_{k=1}^{K}(1+\frac{1}{H})^{h-1}\xi_{h+1}^{k}$$

$$=\sum_{h=1}^{H}\sum_{k=1}^{K}(1+\frac{1}{H})^{h-1}\frac{1}{\check{n}}\sum_{i=1}^{\check{n}}\left(P_{h}^{k}-\mathbf{e}_{s_{h+1}^{\check{l}_{i}}}\right)\left(V_{h+1}^{\check{l}_{i}}-V_{h+1}^{\check{l}_{i},\star}\right)(s_{h}^{k},a_{h}^{k})$$

$$=\sum_{h=1}^{H}\sum_{k=1}^{K}(1+\frac{1}{H})^{h-1}\frac{1}{\check{n}}\sum_{i=1}^{\check{n}}\left(P_{h}^{k}-P_{h}^{\check{l}_{i}}+P_{h}^{\check{l}_{i}}-\mathbf{e}_{s_{h+1}^{\check{l}_{i}}}\right)\left(V_{h+1}^{\check{l}_{i}}-V_{h+1}^{\check{l}_{i},\star}\right)(s_{h}^{k},a_{h}^{k})$$

$$\leq O(KH^{2}\Delta_{p}^{(1)})+\sum_{h=1}^{H}\sum_{k=1}^{K}(1+\frac{1}{H})^{h-1}\frac{1}{\check{n}}\sum_{i=1}^{\check{n}}\left(P_{h}^{\check{l}_{i}}-\mathbf{e}_{s_{h+1}^{\check{l}_{i}}}\right)\left(V_{h+1}^{\check{l}_{i}}-V_{h+1}^{\check{l}_{i},\star}\right)(s_{h}^{k},a_{h}^{k}),\quad(25)$$

where the last step is by the fact that $V_{h+1}^{\check{l}_{i}}(s_{h}^{k},a_{h}^{k})\geq V_{h+1}^{\check{l}_{i},\star}(s_{h}^{k},a_{h}^{k})$ from Lemma 2, and then by Hölder's inequality and the triangle inequality. The following proof is analogous to the proof of Lemma 15 in Zhang et al. (2020). For completeness we reproduce it here. We have

$$\sum_{h=1}^{H}\sum_{k=1}^{K}(1+\frac{1}{H})^{h-1}\frac{1}{\check{n}}\sum_{i=1}^{\check{n}}\left(P_{h}^{\check{l}_{i}}-\mathbf{e}_{s_{h+1}^{\check{l}_{i}}}\right)\left(V_{h+1}^{\check{l}_{i}}-V_{h+1}^{\check{l}_{i},\star}\right)(s_{h}^{k},a_{h}^{k})$$

$$=\sum_{h=1}^{H}\sum_{k=1}^{K}\sum_{j=1}^{K}(1+\frac{1}{H})^{h-1}\frac{1}{\check{n}_{h}^{k}}\sum_{i=1}^{\check{n}_{h}^{k}}\mathbb{1}\left[\check{l}_{h,i}^{k}=j\right]\left(P_{h}^{j}-\mathbf{e}_{s_{h+1}^{j}}\right)\left(V_{h+1}^{j}-V_{h+1}^{j,\star}\right)(s_{h}^{k},a_{h}^{k})$$

$$=\sum_{h=1}^{H}\sum_{k=1}^{K}\sum_{j=1}^{K}(1+\frac{1}{H})^{h-1}\frac{1}{\check{n}_{h}^{k}}\sum_{i=1}^{\check{n}_{h}^{k}}\mathbb{1}\left[\check{l}_{h,i}^{k}=j\right]\left(P_{h}^{j}-\mathbf{e}_{s_{h+1}^{j}}\right)\left(V_{h+1}^{j}-V_{h+1}^{j,\star}\right)(s_{h}^{j},a_{h}^{j}),\quad(26)$$

where (26) holds because $\check{l}_{h,i}^k(s_h^k, a_h^k) = j$ if and only if $j$ is in the previous stage of $k$ and $(s_h^k, a_h^k) = (s_h^j, a_h^j)$. For simplicity of notations, we define $\theta_{h+1}^k \overset{\text{def}}{=} (1 + \frac{1}{H})^{h-1} \sum_{j=1}^K \frac{1}{\check{n}_h^j} \sum_{i=1}^{\check{n}_h^j} \mathbb{1}\left[\check{l}_{h,i}^j = k\right]$.
Then we further have (note that we have swapped the notation of $j$ and $k$)

$$(26) = \sum_{h=1}^H \sum_{k=1}^K \theta_{h+1}^k \left(P_h^k - \mathbf{e}_{s_{h+1}^k}\right) \left(V_{h+1}^k - V_{h+1}^{k,\star}\right)(s_h^k, a_h^k).$$

For $(k, h) \in [K] \times [H]$, let $x_h^k$ denote the number of occurrences of the triple $(s_h^k, a_h^k, h)$ in the current stage. Define $\tilde{\theta}_{h+1}^k \overset{\text{def}}{=} (1 + \frac{1}{H})^{h-1} \frac{\lfloor (1+\frac{1}{H})x_h^k \rfloor}{x_h^k} \leq 3$. Define $\mathcal{K} \overset{\text{def}}{=} \{(k, h) : \theta_{h+1}^k = \tilde{\theta}_{h+1}^k\}$, and $\bar{\mathcal{K}} \overset{\text{def}}{=} \{(k, h) \in [K] \times [H] : \theta_{h+1}^k \neq \tilde{\theta}_{h+1}^k\}$. Then, we have that

$$(26) = \sum_{h=1}^H \sum_{k=1}^K \tilde{\theta}_{h+1}^k \left(P_h^k - \mathbf{e}_{s_{h+1}^k}\right) \left(V_{h+1}^k - V_{h+1}^{k,\star}\right)(s_h^k, a_h^k)$$

$$+ \sum_{h=1}^H \sum_{k=1}^K (\theta_{h+1}^k - \tilde{\theta}_{h+1}^k) \left(P_h^k - \mathbf{e}_{s_{h+1}^k}\right) \left(V_{h+1}^k - V_{h+1}^{k,\star}\right)(s_h^k, a_h^k).$$

Since $\tilde{\theta}_{h+1}^k$ is independent of $s_{h+1}^k$, by the Azuma-Hoeffding inequality, it holds with probability at least $1 - \delta$ that

$$\sum_{h=1}^H \sum_{k=1}^K \tilde{\theta}_{h+1}^k \left(P_h^k - \mathbf{e}_{s_{h+1}^k}\right) \left(V_{h+1}^k - V_{h+1}^{k,\star}\right)(s_h^k, a_h^k) \leq O(\sqrt{KH^3\iota}). \tag{27}$$

It is easy to see that if $k$ is in a stage that is before the second last stage of the triple $(s_h^k, a_h^k, h)$, then $(k, h) \in \mathcal{K}$. For a triple $(s, a, h)$, define $\mathcal{K}_h^\perp(s, a) \overset{\text{def}}{=} \{k \in [K] : k$ is in the second last stage of the triple $(s, a, h)$, $(s_h^k, a_h^k) = (s, a)\}$. We have that

$$\sum_{h=1}^H \sum_{k=1}^K (\theta_{h+1}^k - \tilde{\theta}_{h+1}^k) \left(P_h^k - \mathbf{e}_{s_{h+1}^k}\right) \left(V_{h+1}^k - V_{h+1}^{k,\star}\right)(s_h^k, a_h^k)$$

$$= \sum_{s,a,h} \sum_{k:(k,h)\in\bar{\mathcal{K}}} \mathbb{1}\left[(s_h^k, a_h^k) = (s, a)\right] (\theta_{h+1}^k - \tilde{\theta}_{h+1}^k) \left(P_h^k - \mathbf{e}_{s_{h+1}^k}\right) \left(V_{h+1}^k - V_{h+1}^{k,\star}\right)(s, a)$$

$$= \sum_{s,a,h} (\theta_{h+1}(s, a) - \tilde{\theta}_{h+1}(s, a)) \sum_{k\in\mathcal{K}_h^\perp(s,a)} \left(P_h^k - \mathbf{e}_{s_{h+1}^k}\right) \left(V_{h+1}^k - V_{h+1}^{k,\star}\right)(s, a), \tag{28}$$

where for a fixed triple $(s, a, h)$, we have defined $\theta_{h+1}(s, a) \overset{\text{def}}{=} \theta_{h+1}^k$, for any $k \in \mathcal{K}_h^\perp(s, a)$. Note that $\theta_{h+1}(s, a)$ is well-defined, because $\theta_{h+1}^{k_1} = \theta_{h+1}^{k_2}, \forall k_1, k_2 \in \mathcal{K}_h^\perp(s, a)$. Similarly, let $\tilde{\theta}_{h+1}(s, a) \overset{\text{def}}{=} \tilde{\theta}_{h+1}^k$ for any $k \in \mathcal{K}_h^\perp(s, a)$, and $\tilde{\theta}_{h+1}(s, a)$ is also well-defined. By the Azuma-Hoeffding inequality and a union bound, it holds with probability at least $1 - KH\delta$ that

$$(28) \leq \sum_{s,a,h} O\left(\sqrt{H^2 \left|\mathcal{K}_h^\perp(s, a)\right| \iota}\right)$$

$$= \sum_{s,a,h} O\left(\sqrt{H^2 \check{N}_h^{K+1}(s, a)\iota}\right)$$

$$\leq O\left(\sqrt{SAH^3\iota \sum_{s,a,h} \check{N}_h^{K+1}(s, a)}\right) \tag{29}$$

$$\leq O\left(\sqrt{SAKH^3\iota}\right) \tag{30}$$

where $\check{N}_h^{K+1}(s, a)$ is defined to be the total number of visitations to the triple $(s, a, h)$ over the entire $K$ episodes. (29) is by the Cauchy-Schwartz inequality. (30) holds because by the way stages are defined, for each triple $(s, a, h)$, the length of its last two stages is at most an $O(1/H)$ fraction of the total number of visitations.

Combining (25), (27) and (30) completes the proof. □

## B.5 PROOF OF LEMMA 4

*Proof.* This proof follows a similar structure as the proof of Lemma 2. It should be clear from the way we update $Q_h(s,a)$ that $Q_h^k(s,a)$ is monotonically decreasing in $k$. We now prove $Q_h^{k,\star}(s,a) - 2(H-h+1)b_\Delta \leq Q_h^{k+1}(s,a)$ for all $s,a,h,k$ by induction on $k$. First, it holds for $k=1$ by our initialization of $Q_h(s,a)$. For $k \geq 2$, now suppose $Q_h^{j,\star}(s,a) - 2(H-h+1)b_\Delta \leq Q_h^{j+1}(s,a) \leq Q_h^j(s,a)$ for all $s,a,h$ and $1 \leq j \leq k$. For a fixed triple $(s,a,h)$, we consider the following two cases.

**Case 1:** $Q_h(s,a)$ is updated in episode $k$. Then with probability at least $1 - 2\delta$

$$Q_h^{k+1}(s,a) = \frac{\check{r}_h(s,a)}{\check{N}_h^k(s,a)} + \frac{\check{v}_h(s,a)}{\check{N}_h^k(s,a)} + b_h^k$$

$$\geq \frac{\check{r}_h(s,a)}{\check{n}} + \frac{1}{\check{n}} \sum_{i=1}^{\check{n}} V_{h+1}^{\check{l}_i,\star}(s_{h+1}^{\check{l}_i}) - 2(H-h)b_\Delta + \sqrt{\frac{H^2}{\check{n}}\iota} + \sqrt{\frac{\iota}{\check{n}}} \qquad (31)$$

$$\geq \frac{\check{r}_h(s,a)}{\check{n}} + \frac{1}{\check{n}} \sum_{i=1}^{\check{n}} P_h^{\check{l}_i} V_{h+1}^{\check{l}_i,\star}(s,a) + \sqrt{\frac{\iota}{\check{n}}} - 2(H-h)b_\Delta \qquad (32)$$

$$= \frac{\check{r}_h(s,a)}{\check{n}} + \frac{1}{\check{n}} \sum_{i=1}^{\check{n}} \left( Q_h^{\check{l}_i,\star}(s,a) - r_h^{\check{l}_i}(s,a) \right) + \sqrt{\frac{\iota}{\check{n}}} - 2(H-h)b_\Delta \qquad (33)$$

$$\geq Q_h^{k,\star}(s,a) - b_\Delta - 2(H-h)b_\Delta. \qquad (34)$$

Inequality (31) is by the induction hypothesis that $Q_{h+1}^{\check{l}_i}(s_{h+1}^{\check{l}_i},a) \geq Q_{h+1}^{\check{l}_i,\star}(s_{h+1}^{\check{l}_i},a) - 2(H-h)b_\Delta, \forall a \in \mathcal{A}$, and hence $V_{h+1}^{\check{l}_i}(s_{h+1}^{\check{l}_i}) \geq V_{h+1}^{\check{l}_i,\star}(s_{h+1}^{\check{l}_i}) - 2(H-h)b_\Delta$. Inequality (32) follows from the Azuma-Hoeffding inequality. (33) uses the Bellman optimality equation. Inequality (34) is by the Hoeffding's inequality that $\frac{1}{\check{n}} \left( \sum_{i=1}^{\check{n}} r_h^{\check{l}_i}(s,a) - \check{r}_h(s,a) \right) \leq \sqrt{\frac{\iota}{\check{n}}}$ with high probability, and by Lemma 1 that $Q_h^{\check{l}_i,\star}(s,a) \geq Q_h^{k,\star}(s,a) - b_\Delta$. According to the monotonicity of $Q_h^k(s,a)$, we know that $Q_h^{k,\star}(s,a) - 2(H-h+1)b_\Delta \leq Q_h^{k+1}(s,a) \leq Q_h^k(s,a)$. In fact, we have proved the stronger statement $Q_h^{k+1}(s,a) \geq Q_h^{k,\star}(s,a) - b_\Delta - 2(H-h)b_\Delta$ that will be useful in Case 2 below.

**Case 2:** $Q_h(s,a)$ is not updated in episode $k$. Then there are two possibilities:

1. If $Q_h(s,a)$ has never been updated from episode 1 to episode $k$: It is easy to see that $Q_h^{k+1}(s,a) = Q_h^k(s,a) = \cdots = Q_h^1(s,a) = H - h + 1 \geq Q_h^{k,\star}(s,a) - 2(H-h+1)b_\Delta$ holds.

2. If $Q_h(s,a)$ has been updated at least once from episode 1 to episode $k$: Let $j$ be the index of the latest episode that $Q_h(s,a)$ was updated. Then, from our induction hypothesis and Case 1, we know that $Q_h^{j+1}(s,a) \geq Q_h^{j,\star}(s,a) - b_\Delta - 2(H-h)b_\Delta$. Since $Q_h(s,a)$ has not been updated from episode $j+1$ to episode $k$, we know that $Q_h^{k+1}(s,a) = Q_h^k(s,a) = \cdots = Q_h^{j+1}(s,a) \geq Q_h^{j,\star}(s,a) - b_\Delta - 2(H-h)b_\Delta \geq Q_h^{k,\star}(s,a) - 2(H-h+1)b_\Delta$, where the last inequality holds because of Lemma 1.

A union bound over all time steps completes our proof. □

## B.6 PROOF SKETCH OF THEOREM 2

*Proof sketch.* We only sketch the difference with respect to the proof of Theorem 1 in the main text. The reader should have no difficulty recovering the complete proof by following the same routine as Theorem 1. Specifically, it suffices to investigate the steps that are involved with Lemma 2.

The dynamic regret of the new algorithm in epoch $d = 1$ now can be expressed as

$$\mathcal{R}^{(d)}(\pi, K) = \sum_{k=1}^{K} \left( V_1^{k,*}(s_1^k) - V_1^{k,\pi}(s_1^k) \right) \leq \sum_{k=1}^{K} \left( V_1^k(s_1^k) - V_1^{k,\pi}(s_1^k) \right) + 2KHb_\Delta, \qquad (35)$$

where we applied the results of Lemma 4 instead of Lemma 2. The reader should bear in mind that from the new update rules of the value functions, we now have

$$V_h^k(s_h^k) \leq \mathbb{1}\left[n_h^k = 0\right]H + \frac{\check{r}_h(s_h^k, a_h^k)}{\check{N}_h^k(s_h^k, a_h^k)} + \frac{\check{v}_h(s_h^k, a_h^k)}{\check{N}_h^k(s_h^k, a_h^k)} + b_h^k, \tag{36}$$

where the RHS no longer has the additional bonus term $b_\Delta$. If we define $\zeta_h^k$, $\xi_{h+1}^k$, and $\phi_{h+1}^k$ in the same way as before, the author can easily verify that all the derivations until Equation (12) still holds, although the value of $\Lambda_{h+1}^k$ should be re-defined as $\Lambda_{h+1}^k \stackrel{\text{def}}{=} \xi_{h+1}^k + \phi_{h+1}^k + 3b_h^k + 3b_\Delta$ due to the new upper bound in (36) that is independent of $b_\Delta$. Proposition 1 also follows analogously though some additional attention should be paid to the proof of Lemma 7 where the results of Lemma 2 have been utilized. Finally, we obtain the dynamic regret upper bound in epoch $d = 1$ as follows:

$$\mathcal{R}^{(d)}(\pi, K) \leq \widetilde{O}\left(SAH^3 + \sqrt{SAKH^5} + KH\Delta_r^{(1)} + KH^2\Delta_p^{(1)}\right) + 2KHb_\Delta,$$

where the additional term $2KHb_\Delta$ comes from (35). From our definition of $b_\Delta$, we can easily see that $2KHb_\Delta \leq O(KH\Delta_r^{(1)} + KH^2\Delta_p^{(1)})$. Therefore, we can conclude that the dynamic regret upper bound in one epoch remains the same order, which leaves the dynamic regret over the entire horizon also unchanged. $\qquad\square$

## C  ALGORITHM: RESTARTQ-UCB (FREEDMAN)

The algorithm Restarted Q-Learning with Freedman Upper Confidence Bounds (RestartQ-UCB Freedman) is presented in Algorithm 2. For ease of exposition, we use $\check{r}$, $\check{\mu}$, $\check{v}$, $\check{\sigma}$, $\mu^{\text{ref}}$, $\sigma^{\text{ref}}$, $\check{n}$, and $n$ to denote $\check{r}_h(s, a)$, $\check{\mu}_h(s_h^k, a_h^k)$, $\check{v}_h(s_h^k, a_h^k)$, $\check{\sigma}_h(s_h^k, a_h^k)$, $\mu_h^{\text{ref}}(s_h^k, a_h^k)$, $\sigma_h^{\text{ref}}(s_h^k, a_h^k)$, $\check{N}_h(s_h^k, a_h^k)$, and $N_h(s_h^k, a_h^k)$ respectively, when the values of $(s, a, h, k)$ are clear from the context.

Compared with Algorithm 1, there are two major improvements in Algorithm 2. The first one is to replace the Hoeffding-based bonus term $b_h^k$ with a tighter term $\underline{b}_h^k$. The latter term takes into account the second moment information of the random variables, which allows sharper tail bounds that rely on second moments to come into use (in our case, the Freedman's inequality). The second improvement is a variance reduction technique, or more specifically, the reference-advantage decomposition as coined in Zhang et al. (2020). The intuition is to first learn a reference value function $V^{\text{ref}}$ that serves as a roughly accurate estimate of the optimal value function $V^\star$. The goal of learning the optimal value function $V^\star = V^{\text{ref}} + (V^* - V_{\text{ref}})$ can hence be decomposed into estimating two terms $V^{\text{ref}}$ and $V^* - V_{\text{ref}}$. The reference value $V^{\text{ref}}$ is a fixed term, and can be accurately estimated using a large number of samples (in Algorithm 2, we estimate $V^{\text{ref}}$ only when we have $cSAH^6\iota$ samples for a large constant $c$). The advantage term $V^* - V^{\text{ref}}$ can also be accurately estimated due to the reduced variance.

## D  PROOF OF THEOREM 3

Similar to the proof of Theorem 1, we start with the dynamic regret in one epoch, and then extend to all epochs in the end. The proof follows the same routine as in the proof of Theorem 1. Given that a rigorous analysis on the Freedman-based bonus with variance reduction is present in Zhang et al. (2020), one should not find it difficult to extend our Hoeffding-based algorithm to Algorithm 2. Therefore, rather than providing a complete proof of Theorem 3, in the following, we sketch the differences and highlight the additional analysis needed that is not covered by the proof of Theorem 1 and Zhang et al. (2020).

To facilitate the analysis, first recall a few notations $N_h^k, \check{N}_h^k, Q_h^k(s, a), V_h^k(s), n_h^k, l_{h,i}^k, \check{n}_h^k, \check{l}_{h,i}^k, l_i$ and $\check{l}_i$ that we have defined in Section 4. In addition, when $(h, k)$ is clear from the context, we drop the time indices and simply use $\check{\mu}, \check{\sigma}, \mu^{\text{ref}}, \sigma^{\text{ref}}$ to denote their corresponding values in the computation of the $Q_h(s_h^k, a_h^k)$ value in Line 15 of Algorithm 2.

We start with the following lemma, which is an analogue of Lemma 2 but requires a more careful treatment of variations accumulated in $\mu^{\text{ref}}$ and $\check{\mu}_h$. It states that the optimistic $Q_h^k(s, a)$ is an upper bound of the optimal $Q_h^{k,\star}(s, a)$ with high probability.

---

**Algorithm 2:** RestartQ-UCB (Freedman)

---

1 **for** *epoch* $d \leftarrow 1$ *to* $D$ **do**
2     **Initialize:** $V_h(s) \leftarrow H - h + 1, Q_h(s,a) \leftarrow H - h + 1, N_h(s,a) \leftarrow 0, \check{N}_h(s,a) \leftarrow 0,$
    $\check{r}_h(s,a) \leftarrow 0, \check{\mu}_h(s,a) \leftarrow 0, \check{v}_h(s,a) \leftarrow 0, \check{\sigma}_h(s,a) \leftarrow 0, \mu_h^{\text{ref}}(s,a) \leftarrow 0, \sigma_h^{\text{ref}}(s,a) \leftarrow$
    $0, V_h^{\text{ref}}(s) \leftarrow H,$ for all $(s,a,h) \in \mathcal{S} \times \mathcal{A} \times [H];$
3     **for** *episode* $k \leftarrow (d-1)K + 1$ *to* $\min\{dK, M\}$ **do**
4         observe $s_1^k$;
5         **for** *step* $h \leftarrow 1$ *to* $H$ **do**
6             Take action $a_h^k \leftarrow \arg\max_a Q_h(s_h^k, a)$, receive $R_h^k(s_h^k, a_h^k)$, and observe $s_{h+1}^k$;
7             $\check{r} \leftarrow \check{r} + R_h^k(s_h^k, a_h^k), \check{v} \leftarrow \check{v} + V_{h+1}(s_{h+1}^k);$
8             $\check{\mu} \leftarrow \check{\mu} + V_{h+1}(s_{h+1}^k) - V_{h+1}^{\text{ref}}(s_{h+1}^k), \check{\sigma} \leftarrow \check{\sigma} + \left(V_{h+1}(s_{h+1}^k) - V_{h+1}^{\text{ref}}(s_{h+1}^k)\right)^2;$
9             $\mu^{\text{ref}} \leftarrow \mu^{\text{ref}} + V_{h+1}^{\text{ref}}(s_{h+1}^k), \sigma^{\text{ref}} \leftarrow \sigma^{\text{ref}} + (V_{h+1}^{\text{ref}}(s_{h+1}^k))^2;$
10             $n \leftarrow n + 1, \check{n} \leftarrow \check{n} + 1;$
11             **if** $n \in \mathcal{L}$   // Reaching the end of the stage
12             **then**
13                 $b_h^k \leftarrow \sqrt{\frac{H^2}{\check{n}}\iota} + \sqrt{\frac{1}{\check{n}}\iota}, \ b_\Delta \leftarrow \Delta_r^{(d)} + H\Delta_p^{(d)};$
14                 $\underline{b}_h^k \leftarrow 2\sqrt{\frac{\sigma^{\text{ref}}/n - (\mu^{\text{ref}}/n)^2}{n}\iota} + 2\sqrt{\frac{\check{\sigma}/\check{n} - (\check{\mu}/\check{n})^2}{\check{n}}\iota} + 5(\frac{H\iota}{n} + \frac{H\iota}{\check{n}} + \frac{H\iota^{3/4}}{n^{3/4}} + \frac{H\iota^{3/4}}{\check{n}^{3/4}}) + \sqrt{\frac{1}{\check{n}}\iota};$
15                 $Q_h(s_h^k, a_h^k) \leftarrow \min\left\{\frac{\check{r}}{\check{n}} + \frac{\check{v}}{\check{n}} + b_h^k + 2b_\Delta, \frac{\check{r}}{\check{n}} + \frac{\mu^{\text{ref}}}{n} + \frac{\check{\mu}}{\check{n}} + 2\underline{b}_h^k + 4b_\Delta, Q_h(s_h^k, a_h^k)\right\};$
16                 $V_h(s_h^k) \leftarrow \max_a Q_h(s_h^k, a);$
17                 $\check{N}_h(s_h^k, a_h^k), \check{r}_h(s_h^k, a_h^k), \check{v}_h(s_h^k, a_h^k), \check{\mu}_h(s_h^k, a_h^k), \check{\sigma}_h(s_h^k, a_h^k) \leftarrow 0;$
18                 **if** $\sum_a N_h(s_h^k, a) = \Omega(SAH^6\iota)$// Learn the reference value
19                 **then**
20                     $V_h^{\text{ref}}(s_h^k) \leftarrow V_h(s_h^k);$

---

**Lemma 8.** *(Freedman) For $\delta \in (0,1)$, with probability at least $1 - 2KH\delta$, it holds that $Q_h^{k,\star}(s,a) \leq Q_h^{k+1}(s,a) \leq Q_h^k(s,a), \forall(s,a,h,k) \in \mathcal{S} \times \mathcal{A} \times [H] \times [K]$.*

*Proof.* It should be clear from the way we update $Q_h(s,a)$ that $Q_h^k(s,a)$ is monotonically decreasing in $k$. We now prove $Q_h^{k,\star}(s,a) \leq Q_h^{k+1}(s,a)$ for all $s,a,h,k$ by induction on $k$. First, it holds for $k = 1$ by our initialization of $Q_h(s,a)$. For $k \geq 2$, now suppose $Q_h^{j,\star}(s,a) \leq Q_h^j(s,a)$ for all $s,a,h$ and $1 \leq j \leq k$. For a fixed triple $(s,a,h)$, we consider the following two cases.

**Case 1:** $Q_h(s,a)$ is updated in episode $k$. Notice that it suffices to analyze the case where $Q_h(s,a)$ is updated using $\underline{b}_h^k$, because the other case of $b_h^k$ would be exactly the same as in Lemma 2. With probability at least $1 - \delta$,

$$
\begin{aligned}
Q_h^{k+1}(s,a) &= \frac{\check{r}_h(s,a)}{\check{N}_h^k(s,a)} + \frac{\mu^{\text{ref}}(s,a)}{N_h^k(s,a)} + \frac{\check{\mu}_h(s,a)}{\check{N}_h^k(s,a)} + 2\underline{b}_h^k + 4b_\Delta \\
&= \frac{\check{r}_h(s,a)}{\check{n}} + \underbrace{\frac{1}{n}\sum_{i=1}^n \left(V_{h+1}^{\text{ref},l_i}(s_{h+1}^{l_i}) - P_h^{l_i}V_{h+1}^{\text{ref},l_i}(s,a)\right)}_{\chi_1} \\
&\quad + \underbrace{\frac{1}{\check{n}}\sum_{i=1}^{\check{n}}\left[\left(V_{h+1}^{\check{l}_i}(s_{h+1}^{\check{l}_i}) - V_{h+1}^{\text{ref},\check{l}_i}(s_{h+1}^{\check{l}_i})\right) - \left(P_h^{\check{l}_i}V_{h+1}^{\check{l}_i} - P_h^{\check{l}_i}V_{h+1}^{\text{ref},\check{l}_i}\right)(s,a)\right]}_{\chi_2} \\
&\quad + \underbrace{\frac{1}{n}\sum_{i=1}^n P_h^{l_i}V_{h+1}^{\text{ref},l_i} + \frac{1}{\check{n}}\sum_{i=1}^{\check{n}}\left(P_h^{\check{l}_i}V_{h+1}^{\check{l}_i} - P_h^{\check{l}_i}V_{h+1}^{\text{ref},\check{l}_i}\right)(s,a)}_{\chi_3} + 2\underline{b}_h^k + 4b_\Delta \quad (37)
\end{aligned}
$$

In the following, we will bound each term in (37) separately. First, we have that

$$\chi_3 + 2b_\Delta = \frac{1}{n} \sum_{i=1}^{n} \left( P_h^{l_i} V_{h+1}^{\text{ref},l_i} - P_h^k V_{h+1}^{\text{ref},l_i} \right)(s,a) + b_\Delta \tag{38}$$

$$- \frac{1}{\check{n}} \sum_{i=1}^{\check{n}} \left( P_h^{\check{l}_i} V_{h+1}^{\text{ref},\check{l}_i} - P_h^k V_{h+1}^{\text{ref},\check{l}_i} \right)(s,a) + b_\Delta \tag{39}$$

$$+ \frac{1}{n} \sum_{i=1}^{n} P_h^k V_{h+1}^{\text{ref},l_i}(s,a) - \frac{1}{\check{n}} \sum_{i=1}^{\check{n}} P_h^k V_{h+1}^{\text{ref},\check{l}_i}(s,a) + \frac{1}{\check{n}} \sum_{i=1}^{\check{n}} P_h^{\check{l}_i} V_{h+1}^{\check{l}_i}(s,a) \tag{40}$$

$$\geq \frac{1}{\check{n}} \sum_{i=1}^{\check{n}} P_h^{\check{l}_i} V_{h+1}^{\check{l}_i}(s,a), \tag{41}$$

where (38)$\geq 0$ and (39)$\geq 0$ by Hölder's inequality and the definition of $b_\Delta$. In (40), we have that $\frac{1}{n} \sum_{i=1}^{n} P_h^k V_{h+1}^{\text{ref},l_i}(s,a) - \frac{1}{\check{n}} \sum_{i=1}^{\check{n}} P_h^k V_{h+1}^{\text{ref},\check{l}_i}(s,a) \geq 0$, because $V_{h+1}^{\text{ref},k}(s)$ is non-increasing in $k$.

Following a similar procedure as in Lemma 10, Lemma 12, and Lemma 13 in Zhang et al. (2020), we can further bound $|\chi_1|$ and $|\chi_2|$ as follows:

$$|\chi_1| \leq 2\sqrt{\frac{\nu^{\text{ref}}\iota}{n}} + \frac{5H\iota^{\frac{3}{4}}}{n^{\frac{3}{4}}} + \frac{2\sqrt{\iota}}{Tn} + \frac{2H\iota}{n}, \tag{42}$$

$$|\chi_2| \leq 2\sqrt{\frac{\check{\nu}\iota}{\check{n}}} + \frac{5H\iota^{\frac{3}{4}}}{\check{n}^{\frac{3}{4}}} + \frac{2\sqrt{\iota}}{T\check{n}} + \frac{2H\iota}{\check{n}}, \tag{43}$$

where $\nu^{\text{ref}} \overset{\text{def}}{=} \frac{\sigma^{\text{ref}}}{n} - \left(\frac{\mu^{\text{ref}}}{n}\right)^2$ and $\check{\nu} \overset{\text{def}}{=} \frac{\check{\sigma}}{\check{n}} - \left(\frac{\check{\mu}}{\check{n}}\right)^2$. These are the steps where Freedman's inequality Freedman (1975) come into use, and we omit these steps since they are essentially the same as the derivations in Zhang et al. (2020). We can see from (42), (43), and the definition of $\underline{b}_h^k$ that $|\chi_1| + |\chi_2| \leq \underline{b}_h^k$.

Substituting the results on $\chi_1, \chi_2$ and $\chi_3$ back to (37), it holds that with probability at least $1 - \delta$,

$$Q_h^{k+1}(s,a) = \frac{\check{r}_h(s,a)}{\check{n}} + \chi_1 + \chi_2 + \chi_3 + 2\underline{b}_h^k + 4b_\Delta$$

$$\geq \frac{\check{r}_h(s,a)}{\check{n}} + \frac{1}{\check{n}} \sum_{i=1}^{\check{n}} P_h^{\check{l}_i} V_{h+1}^{\check{l}_i}(s,a) + \underline{b}_h^k + 2b_\Delta \tag{44}$$

$$\geq \frac{\check{r}_h(s,a)}{\check{n}} + \frac{1}{\check{n}} \sum_{i=1}^{\check{n}} P_h^{\check{l}_i} V_{h+1}^{\check{l}_i,\star}(s,a) + \underline{b}_h^k + 2b_\Delta \tag{45}$$

$$= \frac{\check{r}_h(s,a)}{\check{n}} + \frac{1}{\check{n}} \sum_{i=1}^{\check{n}} \left( Q_h^{\check{l}_i,\star}(s,a) - r_h^{\check{l}_i}(s,a) \right) + \underline{b}_h^k + 2b_\Delta$$

$$\geq \frac{1}{\check{n}} \sum_{i=1}^{\check{n}} Q_h^{\check{l}_i,\star}(s,a) + 2b_\Delta \geq Q_h^{k,\star}(s,a) + b_\Delta, \tag{46}$$

where in (44) we used (41), (42), (43), and the definition of $\underline{b}_h^k$ in Algorithm 2. (45) is by the induction hypothesis that $Q_{h+1}^{\check{l}_i}(s_{h+1}^{\check{l}_i}, a) \geq Q_{h+1}^{\check{l}_i,\star}(s_{h+1}^{\check{l}_i}, a), \forall a \in \mathcal{A}, 1 \leq \check{l}_i \leq k$. The second to last inequality holds due to the Hofdding's inequality that $\frac{1}{\check{n}} \left( \sum_{i=1}^{\check{n}} r_h^{\check{l}_i}(s,a) - \check{r}_h(s,a) \right) \leq \sqrt{\frac{\iota}{\check{n}}} \leq \underline{b}_h^k$ with high probability. Finally, the last inequality follows from Lemma 1.

According to the monotonicity of $Q_h^k(s,a)$, we can conclude from (46) that $Q_h^{k,\star}(s,a) \leq Q_h^{k+1}(s,a) \leq Q_h^k(s,a)$. In fact, we have proved the stronger statement $Q_h^{k+1}(s,a) \geq Q_h^{k,\star}(s,a) + b_\Delta$ that will be useful in Case 2 below.

**Case 2:** $Q_h(s,a)$ is not updated in episode $k$. Then, there are two possibilities:

1. If $Q_h(s, a)$ has never been updated from episode 1 to episode $k$: It is easy to see that $Q_h^{k+1}(s, a) = Q_h^k(s, a) = \cdots = Q_h^1(s, a) = H - h + 1 \geq Q_h^{k,\star}(s, a)$ holds.

2. If $Q_h(s, a)$ has been updated at least once from episode 1 to episode $k$: Let $j$ be the index of the latest episode that $Q_h(s, a)$ was updated. Then, from our induction hypothesis and Case 1, we know that $Q_h^{j+1}(s, a) \geq Q_h^{j,\star}(s, a) + b_\Delta$. Since $Q_h(s, a)$ has not been updated from episode $j + 1$ to episode $k$, we know that $Q_h^{k+1}(s, a) = Q_h^k(s, a) = \cdots = Q_h^{j+1}(s, a) \geq Q_h^{j,\star}(s, a) + b_\Delta \geq Q_h^{k,\star}(s, a)$, where the last inequality holds because of Lemma 1.

A union bound over all time steps completes our proof. $\qquad\square$

Conditional on the successful event of Lemma 8, the dynamic regret of Algorithm 2 in epoch $d = 1$ can hence be expressed as

$$\mathcal{R}^{(d)}(\pi, K) = \sum_{k=1}^K \left( V_1^{k,*}\left(s_1^k\right) - V_1^{k,\pi}\left(s_1^k\right) \right) \leq \sum_{k=1}^K \left( V_1^k\left(s_1^k\right) - V_1^{k,\pi}\left(s_1^k\right) \right). \tag{47}$$

From the update rules of the value functions in Algorithm 2, we have

$$V_h^k(s_h^k) \leq \mathbb{1}\left[n_h^k = 0\right] H + \frac{\check{r}_h(s_h^k, a_h^k)}{\check{n}} + \frac{\mu_h^{\mathrm{ref},k}}{n} + \frac{\check{\mu}_h^k}{\check{n}} + 2\underline{b}_h^k + 4b_\Delta$$

$$= \mathbb{1}\left[n_h^k = 0\right] H + \frac{\check{r}_h(s_h^k, a_h^k)}{\check{n}} + \frac{1}{n}\sum_{i=1}^n V_{h+1}^{\mathrm{ref},l_i}(s_{h+1}^{l_i}) + \frac{1}{\check{n}}\sum_{i=1}^{\check{n}}(V_{h+1}^{\check{l}_i}(s_{h+1}^{\check{l}_i}) - V_{h+1}^{\mathrm{ref},\check{l}_i}(s_{h+1}^{\check{l}_i})) + 2\underline{b}_h^k + 4b_\Delta.$$

If we again define $\zeta_h^k \stackrel{\mathrm{def}}{=} V_h^k(s_h^k) - V_h^{k,\pi}(s_h^k)$, we can follow a similar routine as in the proof of Theorem 1 (details can be found in Zhang et al. (2020)) and obtain

$$\sum_{k=1}^K \zeta_1^k \leq O\left(SAH^3 + \sum_{h=1}^H\sum_{k=1}^K (1 + \frac{1}{H})^{h-1}\Lambda_{h+1}^k\right),$$

where $\Lambda_{h+1}^k \stackrel{\mathrm{def}}{=} \psi_{h+1}^k + \xi_{h+1}^k + \phi_{h+1}^k + 4\underline{b}_h^k + 8b_\Delta$ with the following definitions:

$$\psi_{h+1}^k \stackrel{\mathrm{def}}{=} \frac{1}{n_h^k}\sum_{i=1}^{n_h^k} \left(P_h^k V_{h+1}^{\mathrm{ref},l_i} - P_h^k V_{h+1}^{\mathrm{ref},K+1}\right)(s_h^k, a_h^k),$$

$$\xi_{h+1}^k \stackrel{\mathrm{def}}{=} \frac{1}{\check{n}_h^k}\sum_{i=1}^{\check{n}_h^k} \left(P_h^k - \mathbf{e}_{s_{h+1}^{\check{l}_i}}\right)\left(V_{h+1}^{\check{l}_i} - V_{h+1}^{\check{l}_i,\star}\right)(s_h^k, a_h^k),$$

$$\phi_{h+1}^k \stackrel{\mathrm{def}}{=} \left(P_h^k - \mathbf{e}_{s_{h+1}^k}\right)\left(V_{h+1}^{\check{l}_i,\star} - V_{h+1}^{k,\pi}\right)(s_h^k, a_h^k).$$

An upper bound on the first four terms in $\Lambda_{h+1}^k$ is derived in the proof of Lemma 7 in Zhang et al. (2020) (There is an extra term of $\sqrt{\frac{1}{\check{n}}\iota}$ in our defnition of $\underline{b}_h^k$ compared to theirs, but it does not affect the leading term in the upper bound). By further recalling the definition of $b_\Delta$, we can obtain the following lemma.

**Lemma 9.** *(Lemma 7 in Zhang et al. (2020)) With probability at least $(1 - O(H^2T^4\delta))$, it holds that*

$$\sum_{h=1}^H\sum_{k=1}^K (1 + \frac{1}{H})^{h-1}\Lambda_{h+1}^k = O\left(\sqrt{SAH^2T\iota} + H\sqrt{T\iota}\log(T) + S^2 A^{\frac{3}{2}} H^8 T^{\frac{1}{4}}\iota + KH\Delta_r^{(1)} + KH^2\Delta_p^{(1)}\right).$$

Combined with (47) and the definition of $\zeta_h^k$, we obtain the dynamic regret bound in a single epoch:

$$\mathcal{R}^{(d)}(\pi, K) = O\left(\sqrt{SAH^2T\iota} + H\sqrt{T\iota}\log(T) + S^2 A^{\frac{3}{2}} H^8 T^{\frac{1}{4}}\iota + KH\Delta_r^{(1)} + KH^2\Delta_p^{(1)}\right), \forall d \in [D].$$

Finally, suppose $T$ is greater than a polynomial of $S, A, \Delta$ and $H$, $\sqrt{SAH^2T\iota}$ would be the leading term of the dynamic regret in a single epoch. In this case, summing up the dynamic regret over all the $D$ epochs gives us an upper bound of $\widetilde{O}\left(D\sqrt{SAH^2T} + \sum_{d=1}^{D} KH\Delta_r^{(d)} + \sum_{d=1}^{D} KH^2\Delta_p^{(d)}\right)$. Recall that $\sum_{d=1}^{D}\Delta_r^{(d)} \le \Delta_r$, $\sum_{d=1}^{D}\Delta_p^{(d)} \le \Delta_p$, $\Delta = \Delta_r + \Delta_p$, and that $K = \Theta(\frac{T}{DH})$. By setting $D = S^{-\frac{1}{3}}A^{-\frac{1}{3}}\Delta^{\frac{2}{3}}T^{\frac{1}{3}}$, the dynamic regret over the entire $T$ steps is bounded by

$$\mathcal{R}(\pi, M) \le \widetilde{O}\left(S^{\frac{1}{3}}A^{\frac{1}{3}}\Delta^{\frac{1}{3}}HT^{\frac{2}{3}}\right).$$

This completes the proof of Theorem 3.

## E   PROOF OF THEOREM 4

The proof of our lower bound relies on the construction of a "hard instance" of non-stationary MDPs. The instance we construct is essentially a switching-MDP: an MDP with piecewise constant dynamics on each *segment* of the horizon, and its dynamics experience an abrupt change at the beginning of each new segment. More specifically, we divide the horizon $T$ into $L$ segments[3], where each segment has $T_0 \stackrel{\text{def}}{=} \lfloor \frac{T}{L} \rfloor$ steps and contains $M_0 \stackrel{\text{def}}{=} \lfloor \frac{M}{L} \rfloor$ episodes, each episode having a length of $H$. Within each such segment, the system dynamics of the MDP do not vary, and we construct the dynamics for each segment in a way such that the instance is a hard instance of stationary MDPs on its own. The MDP within each segment is essentially similar to the hard instances constructed in stationary RL problems (Osband & Van Roy, 2016; Jin et al., 2018). Between two consecutive segments, the dynamics of the MDP change abruptly, and we let the dynamics vary in a way such that no information learned from previous interactions with the MDP can be used in the new segment. In this sense, the agent needs to learn a new hard stationary MDP in each segment. Finally, optimizing the value of $L$ and the variation magnitude between consecutive segments (subject to the constraints of the total variation budget) leads to our lower bound.

We start with a simplified episodic setting where the transition kernels and reward functions are held constant within each episode, i.e., $P_1^m = \cdots = P_h^m = \ldots P_H^m$ and $r_1^m = \cdots = r_h^m = \ldots r_H^m, \forall m \in [M]$. This is a popular but less challenging episodic setting, and its stationary counterpart has been studied in Azar et al. (2017). We further require that when the environment varies due to the non-stationarity, all steps in one episode should vary simultaneously in the same way. This simplified setting is easier to analyze, and its analysis conveniently leads to a lower bound for the un-discounted setting as a side result along the way. Later we will show how the analysis can be naturally extended to the more general setting we introduced in Section 2, using techniques that have also been utilized in Jin et al. (2018). For simplicity of notations, we temporarily drop the $h$ indices and use $P^m$ and $r^m$ to denote the transition kernel and reward function whenever there is no ambiguity.

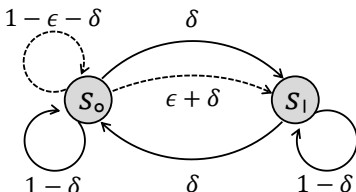

Figure 1: The "JAO MDP" constructed in Jaksch et al. (2010). Dashed lines denote transitions related to the good action $a^\star$.

Consider a two-state MDP as depicted in Figure 1. This MDP was initially proposed in Jaksch et al. (2010) as a hard instance of stationary MDPs, and following Jin et al. (2018) we will refer to this construction as the "JAO MDP". This MDP has 2 states $\mathcal{S} = \{s_\circ, s_|\}$ and $SA$ actions $\mathcal{A} = \{1, 2, \ldots, SA\}$. The reward does not depend on actions: state $s_|$ always gives reward 1 whatever action is taken, and state $s_\circ$ always gives reward 0. Any action taken at state $s_|$ takes the agent to state $s_\circ$ with probability $\delta$, and to state $s_|$ with probability $1 - \delta$. At state $s_\circ$, for all but a single

---

[3]The definition of segments is irrelevant to, and should not be confused with, the notion of epochs we previously defined.

"good" action $a^\star$, the agent is taken to state $s_\mathsf{l}$ with probability $\delta$, and for the good action $a^\star$, the agent is taken to state $s_\mathsf{l}$ with probability $\delta + \varepsilon$ for some $0 < \varepsilon < \delta$. The exact values of $\delta$ and $\varepsilon$ will be chosen later. Note that this is not an MDP with $S$ states and $A$ actions as we desire, but the extension to an MDP with $S$ states and $A$ actions is routine (Jaksch et al., 2010), and is hence omitted here.

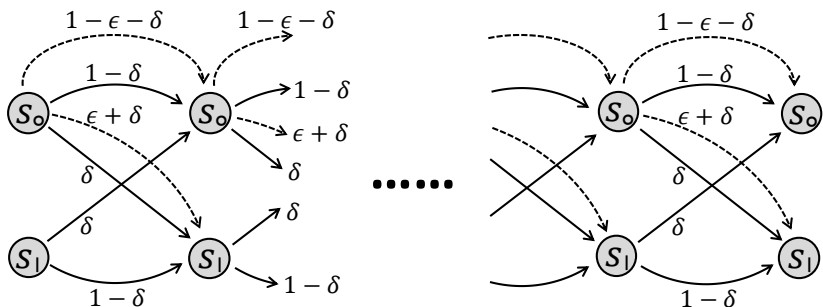

Figure 2: A chain with $H$ copies of JAO MDPs correlated in time. At the end of an episode, the state should deterministically transition from any state in the last copy to the $s_\circ$ state in the first copy of the chain, the arrows of which are not shown in the figure. Also, the $s_\mathsf{l}$ state in the first copy is actually never reached and hence is redundant.

To apply the JAO MDP to the simplified episodic setting, we "concatenate" $H$ copies of exactly the same JAO MDP into a chain as depicted in Figure 2, denoting the $H$ steps in an episode. The initial state of this MDP is the $s_\circ$ state in the first copy of the chain, and after each episode the state is "reset" to the initial state. In the following, we first show that the constructed MDP is a hard instance of stationary MDPs, without worrying about the evolution of the system dynamics. The techniques that we will be using are essentially the same as in the proofs of the lower bound in the multi-armed bandit problem (Auer et al., 2002) or the reinforcement learning problem in the un-discounted setting (Jaksch et al., 2010).

The good action $a^\star$ is chosen uniformly at random from the action space $\mathcal{A}$, and we use $\mathbb{E}_\star[\cdot]$ to denote the expectation with respect to the random choice of $a^\star$. We write $\mathbb{E}_a[\cdot]$ for the expectation conditioned on action $a$ being the good action $a^\star$. Finally, we use $\mathbb{E}_{\mathrm{unif}}[\cdot]$ to denote the expectation when there is no good action in the MDP, i.e., every action in $\mathcal{A}$ takes the agent from state $s_\circ$ to $s_\mathsf{l}$ with probability $\delta$. Define the probability notations $\mathbb{P}_\star(\cdot)$, $\mathbb{P}_a(\cdot)$, and $\mathbb{P}_{\mathrm{unif}}(\cdot)$ analogously.

Consider running a reinforcement learning algorithm on the constructed MDP for $T_0$ steps, where $T_0 = M_0 H$. It has been shown in Auer et al. (2002) and Jaksch et al. (2010) that it is sufficient to consider deterministic policies. Therefore, we assume that the algorithm maps deterministically from a sequence of observations to an action $a_t$ at time $t$. Define the random variables $N_\mathsf{l}$, $N_\circ$ and $N_\circ^\star$ to be the total number of visits to state $s_\mathsf{l}$, the total number of visits to $s_\circ$, and the total number of times that $a^\star$ is taken at state $s_\circ$, respectively. Let $s_t$ denote the state observed at time $t$, and $a_t$ the action taken at time $t$. When there is no chance of ambiguity, we sometimes also use $s_h^m$ to denote the state at step $h$ of episode $m$, which should be interpreted as the state $s_t$ observed at time $t = (m-1) \times H + h$. The notation $a_h^m$ is used analogously. Since $s_\circ$ is assumed to be the initial state, we have that

$$\mathbb{E}_a[N_\mathsf{l}] = \sum_{t=1}^{T_0} \mathbb{P}_a(s_t = s_\mathsf{l}) = \sum_{m=1}^{M_0} \sum_{h=2}^{H} \mathbb{P}_a(s_h^m = s_\mathsf{l})$$

$$= \sum_{m=1}^{M_0} \sum_{h=2}^{H} \left( \mathbb{P}_a(s_{h-1}^m = s_\circ) \cdot \mathbb{P}_a(s_h^m = s_\mathsf{l} \mid s_{h-1}^m = s_\circ) + \mathbb{P}_a(s_{h-1}^m = s_\mathsf{l}) \cdot \mathbb{P}_a(s_h^m = s_\mathsf{l} \mid s_{h-1}^m = s_\mathsf{l}) \right)$$

$$= \sum_{m=1}^{M_0} \sum_{h=2}^{H} \left( \delta \mathbb{P}_a(s_{h-1}^m = s_\circ, a_h^m \neq a^\star) + (\delta + \varepsilon) \mathbb{P}_a(s_{h-1}^m = s_\circ, a_h^m = a^\star) + (1 - \delta) \mathbb{P}_a(s_{h-1}^m = s_\mathsf{l}) \right)$$

$$\leq \delta \mathbb{E}_a[N_\circ - N_\circ^\star] + (\delta + \varepsilon) \mathbb{E}_a[N_\circ^\star] + (1 - \delta) \mathbb{E}_a[N_\mathsf{l}],$$

and rearranging the last inequality gives us $\mathbb{E}_a[N_\mathsf{l}] \leq \mathbb{E}_a[N_\mathsf{o} - N_\mathsf{o}^\star] + (1 + \frac{\varepsilon}{\delta})\mathbb{E}_a[N_\mathsf{o}^\star]$.

For this proof only, define the random variable $W(T_0)$ to be the total reward of the algorithm over the horizon $T_0$, and define $G(T_0)$ to be the (static) regret with respect to the optimal policy. Since for any algorithm, the probability of staying in state $s_\mathsf{o}$ under $\mathbb{P}_a(\cdot)$ is no larger than under $\mathbb{P}_{\mathrm{unif}}(\cdot)$, it follows that

$$\mathbb{E}_a[W(T_0)] \leq \mathbb{E}_a[N_\mathsf{l}] \leq \mathbb{E}_a[N_\mathsf{o} - N_\mathsf{o}^\star] + (1 + \frac{\varepsilon}{\delta})\mathbb{E}_a[N_\mathsf{o}^\star]$$

$$= \mathbb{E}_a[N_\mathsf{o}] + \frac{\varepsilon}{\delta}\mathbb{E}_a[N_\mathsf{o}^\star] \leq \mathbb{E}_{\mathrm{unif}}[N_\mathsf{o}] + \frac{\varepsilon}{\delta}\mathbb{E}_a[N_\mathsf{o}^\star]$$

$$= T_0 - \mathbb{E}_{\mathrm{unif}}[N_\mathsf{l}] + \frac{\varepsilon}{\delta}\mathbb{E}_a[N_\mathsf{o}^\star]. \tag{48}$$

Let $\tau_{\mathsf{ol}}^m$ denote the first step that the state transits from state $s_\mathsf{o}$ to $s_\mathsf{l}$ in the $m$-th episode, then

$$\mathbb{E}_{\mathrm{unif}}[N_\mathsf{l}] = \sum_{m=1}^{M_0}\sum_{h=1}^{H}\mathbb{P}_{\mathrm{unif}}(\tau_{\mathsf{ol}}^m = h)\mathbb{E}_{\mathrm{unif}}[N_\mathsf{l} \mid \tau_{\mathsf{ol}}^m = h] = \sum_{m=1}^{M_0}\sum_{h=1}^{H}(1-\delta)^{h-1}\delta\mathbb{E}_{\mathrm{unif}}[N_\mathsf{l} \mid \tau_{\mathsf{ol}}^m = h]$$

$$\geq \sum_{m=1}^{M_0}\sum_{h=1}^{H}(1-\delta)^{h-1}\delta\frac{H-h}{2} = \sum_{m=1}^{M_0}\left(\frac{H}{2} - \frac{1}{2\delta} + \frac{(1-\delta)^H}{2\delta}\right)$$

$$\geq \frac{T_0}{2} - \frac{M_0}{2\delta}. \tag{49}$$

Since the algorithm is a deterministic mapping from the observation sequence to an action, the random variable $N_\mathsf{o}^\star$ is also a function of the observations up to time $T$. In addition, since the immediate reward only depends on the current state, $N_\mathsf{o}^\star$ can further be considered as a function of just the state sequence up to $T$. Therefore, the following lemma from Jaksch et al. (2010), which in turn was adapted from Lemma A.1 in Auer et al. (2002), also applies in our setting.

**Lemma 10.** *(Lemma 13 in Jaksch et al. (2010)) For any finite constant $B$, let $f : \{s_\mathsf{o}, s_\mathsf{l}\}^{T_0+1} \to [0, B]$ be any function defined on the state sequence $\mathbf{s} \in \{s_\mathsf{o}, s_\mathsf{l}\}^{T_0+1}$. Then, for any $0 < \delta \leq \frac{1}{2}$, any $0 < \varepsilon \leq 1 - 2\delta$, and any $a \in \mathcal{A}$, it holds that*

$$\mathbb{E}_a[f(\mathbf{s})] \leq \mathbb{E}_{unif}[f(\mathbf{s})] + \frac{B}{2} \cdot \frac{\varepsilon}{\sqrt{\delta}}\sqrt{2\mathbb{E}_{unif}[N_\mathsf{o}^\star]}.$$

Since $N_\mathsf{o}^\star$ itself is a function from the state sequence to $[0, T_0]$, we can apply Lemma 10 and arrive at

$$\mathbb{E}_a[N_\mathsf{o}^\star] \leq \mathbb{E}_{\mathrm{unif}}[N_\mathsf{o}^\star] + \frac{T_0}{2} \cdot \frac{\varepsilon}{\sqrt{\delta}}\sqrt{2\mathbb{E}_{\mathrm{unif}}[N_\mathsf{o}^\star]}. \tag{50}$$

From (49), we have that $\sum_{a=1}^{SA}\mathbb{E}_{\mathrm{unif}}[N_\mathsf{o}^\star] = T_0 - \mathbb{E}_{\mathrm{unif}}[N_\mathsf{l}] \leq \frac{T_0}{2} + \frac{M_0}{2\delta}$. By the Cauchy-Schwarz inequality, we further have that $\sum_{a=1}^{SA}\sqrt{2\mathbb{E}_{\mathrm{unif}}[N_\mathsf{o}^\star]} \leq \sqrt{SA(T_0 + \frac{M_0}{\delta})}$. Therefore, from (50), we obtain

$$\sum_{a=1}^{SA}\mathbb{E}_a[N_\mathsf{o}^\star] \leq \frac{T_0}{2} + \frac{M_0}{2\delta} + \frac{T_0}{2} \cdot \frac{\varepsilon}{\sqrt{\delta}}\sqrt{SA(T_0 + \frac{M_0}{\delta})}.$$

Together with (48) and (49), it holds that

$$\mathbb{E}_\star[W(T_0)] \leq \frac{1}{SA}\sum_{a=1}^{SA}\mathbb{E}_a[W(T_0)]$$

$$\leq \frac{T_0}{2} + \frac{M_0}{2\delta} + \frac{\varepsilon}{\delta}\frac{1}{SA}\left(\frac{T_0}{2} + \frac{M_0}{2\delta} + \frac{T_0}{2} \cdot \frac{\varepsilon}{\sqrt{\delta}}\sqrt{SA(T_0 + \frac{M_0}{\delta})}\right). \tag{51}$$

### E.1 UN-DISCOUNTED SETTING

Let us now momentarily deviate from the episodic setting and consider the un-discounted setting (with $M_0 = 1$). This is the case of the JAO MDP in Figure 1 where there is not reset. We could

calculate the stationary distribution and find that the optimal average reward for the JAO MDP is $\frac{\delta+\varepsilon}{2\delta+\varepsilon}$. It is also easy to calculate that the diameter of the JAO MDP is $D = \frac{1}{\delta}$. Therefore, the expected (static) regret with respect to the randomness of $a^*$ can be lower bounded by

$$
\begin{aligned}
\mathbb{E}_\star[G(T_0)] =& \frac{\delta+\varepsilon}{2\delta+\varepsilon}T_0 - \mathbb{E}_\star[W(T_0)] \\
\geq& \frac{\varepsilon T_0}{4\delta+2\varepsilon} - \frac{D}{2} - \frac{\varepsilon D(T_0+D)}{2SA} - \frac{\varepsilon^2 T_0 D\sqrt{D}}{2\sqrt{SA}}(\sqrt{T_0}+\sqrt{D}).
\end{aligned}
$$

By assuming $T_0 \geq DSA$ (which in turn suggests $D \leq \sqrt{\frac{T_0 D}{SA}}$) and setting $\varepsilon = c\sqrt{\frac{SA}{T_0 D}}$ for $c = \frac{3}{40}$, we further have that

$$
\begin{aligned}
\mathbb{E}_\star[G(T_0)] \geq& \left( \frac{c}{6} - \frac{c}{2SA} - \frac{cD}{2SAT_0} - \frac{c^2}{2} - \frac{c^2}{2}\sqrt{\frac{D}{T_0}} \right)\sqrt{SAT_0 D} - \frac{D}{2} \\
\geq& \left( \frac{3}{20}c - c^2 - \frac{1}{200} \right)\sqrt{SAT_0 D} = \frac{1}{1600}\sqrt{SAT_0 D}.
\end{aligned}
$$

It is easy to verify that our choice of $\delta$ and $\varepsilon$ satisfies our assumption that $0 < \varepsilon < \delta$. So far, we have recovered the (static) regret lower bound of $\Omega(\sqrt{SAT_0 D})$ in the un-discounted setting, which was originally proved in Jaksch et al. (2010).

Based on this result, let us now incorporate the non-stationarity of the MDP and derive a lower bound for the dynamic regret $\mathcal{R}(T)$. Recall that we are constructing the non-stationary environment as a switching-MDP. For each segment of length $T_0$, the environment is held constant, and the regret lower bound for each segment is $\Omega(\sqrt{SAT_0 D})$. At the beginning of each new segment, we uniformly sample a new action $a^*$ at random from the action space $\mathcal{A}$ to be the good action for the new segment. In this case, the learning algorithm cannot use the information it learned during its previous interactions with the environment, even if it knows the switching structure of the environment. Therefore, the algorithm needs to learn a new (static) MDP in each segment, which leads to a dynamic regret lower bound of $\Omega(L\sqrt{SAT_0 D}) = \Omega(\sqrt{SATLD})$, where let us recall that $L$ is the number of segments. Every time the good action $a^*$ varies, it will cause a variation of magnitude $2\varepsilon$ in the transition kernel. The constraint of the overall variation budget requires that $2\varepsilon L = \frac{3}{20}\sqrt{\frac{SA}{T_0 D}}L \leq \Delta$, which in turn requires $L \leq 4\Delta^{\frac{2}{3}}T^{\frac{1}{3}}D^{\frac{1}{3}}S^{-\frac{1}{3}}A^{-\frac{1}{3}}$. Finally, by assigning the largest possible value to $L$ subject to the variation budget, we obtain a dynamic regret lower bound of $\Omega\left( S^{\frac{1}{3}}A^{\frac{1}{3}}\Delta^{\frac{1}{3}}D^{\frac{2}{3}}T^{\frac{2}{3}} \right)$. This completes the proof of Proposition 2.

### E.2 EPISODIC SETTINGS

Now let us go back to our simplified episodic setting, as depicted in Figure 2. One major difference with the previous un-discounted setting is that we might not have time to mix between $s_\circ$ and $s_\mathsf{l}$ in $H$ steps. (Note that we only need to reach the stationary distribution over the $(s_\circ, s_\mathsf{l})$ pair in each step $h$, rather than the stationary distribution over the entire MDP. In fact, the latter case is never possible because the entire MDP is not aperiodic.) It can be shown that the optimal policy on this MDP has a mixing time of $\Theta\left(\frac{1}{\delta}\right)$ (Jin et al., 2018), and hence we can choose $\delta$ to be slightly larger than $\Theta(\frac{1}{H})$ to guarantee sufficient time to mix. All the analysis up to inequality (51) carries over to the episodic setting, and essentially we can set $\delta$ to be $\Theta\left(\frac{1}{H}\right)$ to get a (static) regret lower bound of $\Omega(\sqrt{SAT_0 H})$ in each segment. Another difference with the previous setting lies in the usage of the variation budget. Since we require that all the steps in the same episode should vary simultaneously, it now takes a variation budget of $2\varepsilon H$ each time we switch to a new action $a^*$ at the beginning of a new segment. Therefore, the overall variation budget now puts a constraint of $2\varepsilon HL \leq O(\Delta)$ on the magnitude of each switch. Again, by choosing $\varepsilon = \Theta\left(\sqrt{\frac{SA}{T_0 H}}\right)$ and optimizing over possible values of $L$ subject to the budget constraint, we obtain a dynamic regret lower bound of $\Omega\left( S^{\frac{1}{3}}A^{\frac{1}{3}}\Delta^{\frac{1}{3}}H^{\frac{1}{3}}T^{\frac{2}{3}} \right)$ in the simplified episodic setting.

Finally, we consider the standard episodic setting as introduced in Section 2. In this setting, we essentially will be concatenating $H$ distinct JAO MDPs, each with an independent good action $a^*$,

into a chain like Figure 2. The transition kernels in these JAO MDPs are also allowed to vary asynchronously in each step $h$, although our construction of the lower bound does not make use of this property. As argued similarly in Jin et al. (2018), the number of observations for each specific JAO MDP is only $T_0/H$, instead of $T_0$. Therefore, we can assign a slightly larger value to $\varepsilon$ and the learning algorithm would still not be able to identify the good action given the fewer observations. Setting $\delta = \Theta\left(\frac{1}{H}\right)$ and $\varepsilon = \Theta\left(\sqrt{\frac{SA}{T_0}}\right)$ leads to a (static) regret lower bound of $\Omega(H\sqrt{SAT_0})$ in the stationary RL problem. Again, the transition kernels in all the $H$ JAO MDPs vary simultaneously at the beginning of each new segment. By optimizing $L$ subject to the overall budget constraint $2\varepsilon H L \leq O(\Delta)$, we obtain a dynamic regret lower bound of $\Omega\left(S^{\frac{1}{3}}A^{\frac{1}{3}}\Delta^{\frac{1}{3}}H^{\frac{2}{3}}T^{\frac{2}{3}}\right)$ in the episodic setting. This completes our proof of Theorem 4.

