# OpenReview forum: "Near-Optimal Regret Bounds for Model-Free RL in Non-Stationary Episodic MDPs"
_ICLR.cc/2021/Conference — Reject_

### Official Review · AnonReviewer2 · 2020-10-19
**An interesting paper, though the assumption is slightly strong**

**Rating:** 7
**Confidence:** 4

**Review:**

The paper studies efficient model-free reinforcement learning in non-stationary Markov Decision process. They propose an algorithm with efficient dynamic regret bounds. Besides, they also propose matching lower bounds.

pros:
- The theoretical results are solid since they achieve minimax regret bounds.

cons:
- The algorithm follows the similar algorithmic framework of [1], which studies provably efficient reinforcement learning with low switching condition. The algorithm of [1] suits non-stationary setting well since it periodically forgets the previous experience and only uses data collected recently. I guess this is the reason why this idea works for non-stationary environment. Maybe more discussion is needed in the paper.

- The assumption about prior knowledge of the variation budget in each epoch is a bit strong. Restart-UCB algorithm require this knowledge to construct the confidence bonus $b_{\Delta}$. However, in real applications, it is almost impossible to know the variation budget beforehand (If we know this, we can almost directly know the reward and transition in each step.) For theoretical analysis, I remember that many previous results don't need this assumptions, including papers about non-stationary MDP and non-stationary bandit. Since the reasonability of this assumption is important to quantify the contribution of the paper, I suggest that the authors should discuss in detail about this issue during the rebuttal period, including whether this assumption is common in the previous literature, and whether it is possible to remove the assumption.

[1] Provably Efficient Q-Learning with Low Switching Cost


-------Post Rebuttal-------

Thanks for the feedback from the authors. I am glad to see that the new theorem (Theorem 2) has successfully tackle the problem about the prior knowledge of the variation budget. I have updated the score accordingly.

---

> ### Author Response · Authors · 2020-11-20
> **Response to Reviewer 2**
>
> We thank the reviewer for the helpful feedback. Our detailed responses are as follows.
>
> 1. We thank the reviewer for suggesting the reference on low switching cost RL. We have included a discussion about it in our revised paper accordingly.
>
> 2. We would like to let the reviewer know that we have successfully removed the assumption about knowledge of the variation budget in each epoch, without any loss on the regret bound. This result is formally presented in the (new) Theorem 2 of the revised paper. This is achieved by a very lightweight modification to the algorithm. Specifically, we replace the Q-value update rule in Equation (∗) of Algorithm 1 with the new update rule in Equation (1). To understand why this simple modification works, notice that in (∗) we are adding exactly the same bonus term $2 b_\Delta$ to the upper confidence bounds of all $(s, a)$ pairs in the same epoch. Subtracting the same value from all optimistic Q-values simultaneously should not change the choice of actions in future steps. This modification incurs some additional complexity in the analysis, because the new “optimistic” $Q^k_h(s, a)$ values would no longer be strict upper bounds of the optimal $Q^k_h(s,a)$ anymore, but only an “upper bound” subject to some error term of the order $b_\Delta$. To handle this, we provide a new analysis on how this error term propagates over time in the (new) Lemma 4. We would also like to remark that the easy removal of the local budget assumption is non-trivial in the design of the algorithm, and to our best knowledge this result does not exist in the non-stationary RL literature with restarts. For example, in a paper (Zhou et al., 2020) posted after our ICLR submission, the proposed algorithm suffers a severe regret degeneration after removing this assumption. This hence can be considered as another contribution of our work.
>
> We really appreciate the enlightening comments from the reviewer, and hope the additional results may improve the appraisal of our paper.

---

### Official Review · AnonReviewer1 · 2020-10-28
**Interesting model but the contribution is unclear to me**

**Rating:** 4
**Confidence:** 4

**Review:**

This paper derives a low-regret algorithm to control non-stationary MDPs. By non-stationary, the authors mean that the transition matrix and reward function can vary over time. The sum of all changes is bounded by a variation budget \Delta (known by the controller). This algorithm combines two ingredients:
1. The learning horizon is decomposed into epochs whose lengths depends on \Delta.
2. Within an epoch, the algorithm uses an unmodified Q-learning UCB.

The authors derive an upper bound on the regret of the algorithm that is obtained by carefully choosing the epoch lengths.

I found the paper reasonably well written (although the technical part are not easy to understand, see below). The algorithm is clearly defined and the results are easy to understand.  The problem of learning in non-stationary environment is quite challenging and hence the result is appealing. The proof approach is quite classical for this kind of problems and looks reasonable (apart from a term of Proposition 1, see below).

That being said, I found that the paper has a number of shortcomings that prevent me from seeing its contributions as remarkable. In particular:

1. What is the main originality of the paper? The idea of restarting for non-stationary environment is not new (Jaksch et al. 2010). The use of Q-learning UCB is not new (Jin et al. 2018). The combination of the two and its analysis is probably new.  Yet, I fail to see it as a real contributions. In particular, I have the impression that the approach taken by the authors could be easily adapted by replacing Q-learning UCB by any other algorithm (UCRL2, PSRL,...).  The usual approach to obtain a regret bound is to view the regret as a term of opportunism + a term of concentration. Here, the only addition is to add a term of "error" due to \Delta and then to tune the epoch lengths. This approach seems algorithm-independent.

2. I do not understand why *model-free* is so important here. The fact that there is no analysis of model-free non-stationary algorithms in the literature might just be because model model-based algorithms perform better? The authors do not compare their algorithms to existing solutions. Some numerical experiments could be useful.

3. On page 8: where did the SAH^3 go? Naively replacing D by its value and T by SA\Delta H^2 leads to a term TH which grows in T. Where is the catch?

4. The heavy notations make the technical parts hard to follow. Also, on the beginning of page 6, the notations \v{n}^k_h are then transformed in \v{n} while being similar to N and \v{N}. I would hope that some simplification are doable.

---

> ### Author Response · Authors · 2020-11-20
> **(2/2) Response to Reviewer 1**
>
> (2/2)
>
> - Significance of model-freeness: In our opinion, no model-free non-stationary algorithm exists in the literature mostly because model-based algorithms are relatively easier to analyze and the analytic techniques (e.g., extended value iteration) are better established for model-based solutions. It is hence relatively  easier to achieve tight regret or sample complexity results using model-based solutions than model-free ones. This can be seen from the historical fact that even in stationary RL, the first near-optimal regret using model-free algorithms (Jin et al., 2018) was achieved much later than by model-based algorithms (Jaksch et al., 2010). However, even with this intuition, as can be seen from Table 1, the existing model-based solutions still perform worse than our near-optimal model-free one. Thus, we believe establishing near-optimal regret bounds for model-free solutions is technically more challenging and non-trivial (why we studied it). Moreover, model-free solutions are generally believed to be simpler, more time- and space-efficient, more flexible to use, and thus more prevalent in modern deep RL architectures. We hence believe that model-free solutions are important goals to pursue in RL research, which has not been investigated in the non-stationary RL world yet (another reason why we studied it). Our work makes an initial attempt, and manages to make it near-optimal. We also agree that numerical experiments could be useful to better illustrate this point. We have been working on the experiments since the rebuttal period started, and we will post the results hopefully by the end of the rebuttal period.
>
> - On page 8: The $SAH^3$ term disappears simply because we have assumed $T = \Omega(SA\Delta H^2)$, and hence $SAH^3$ would not be a dominant term compared with the other terms. It is common in RL regret analysis (e.g., Jin et al., 2018, Zhang et al., 2020) to assume $T$ is sufficiently large so that one can focus on the dominant terms.
>
> - We thank the reviewer for pointing out the issue with notations. We agree that our notations might look a bit heavy, although hopefully our analytic framework is still clear to the readers. We will try to simplify the notation system in the next revision.
>
> We sincerely thank the reviewer again for the helpful comments. Hope our updated version and the rebuttal have cleared all your concerns. We are also always open to further discussions on these technical details.

---

> ### Author Response · Authors · 2020-11-20
> **(1/2) Response to Reviewer 1**
>
> (1/2)
>
> We thank the reviewer for the helpful feedback. Our detailed responses are as follows.
>
> First, we would like to refer the reviewer to the separate “General Responses” regarding an *emphasis* on the contributions of this paper. We have also successfully removed a major assumption on knowledge of variation budgets in the revised paper. The details can be found in the (new) Theorem 2 of the revised paper and “General Responses” as well.
>
> - Originality: As we highlighted in “General Responses”, our main objective in this paper was to investigate the *tightest possible* regret bound in non-stationary RL, and pursue *minimax-optimal* regret as an ultimate goal. Our primary intention was not to propose a completely new strategy to handle non-stationarity, but rather to study the lowest achievable regret in non-stationary RL and its fundamental limits, using techniques that are preferably as simple as possible. We believe that our main contribution should be the *first nearly-tight regret bound* in non-stationary MDPs. Such a tight bound was previously not established in the literature, as summarized in our Table 1, and not even in the relevant papers (Zhou et al., 2020, Touati et al., 2020) that were posted after our ICLR submission. We would like to argue that using simple and standard techniques does not mean that our results/contributions are not novel.
>
> - We would also like to point out that our approach and results are *not algorithm-independent*. As the reviewer has mentioned, applying the restarting strategy to other algorithms (e.g. UCRL2) might lead to *some* sublinear regret bound, but it requires a careful design of the algorithmic framework and non-trivial analysis to achieve a regret bound *as tight as ours*. This is evidenced by the numerous existing efforts devoted to non-stationary RL (cf. Table 1) that failed to achieve a near-optimal regret bound. In fact, in two papers (Zhou et al., 2020, Touati et al., 2020) that were posted even after our ICLR submission, it can be seen that the regret bound is far from tight in their settings (e.g., at an order of $O(d^{1 / 2} H)$) when a restarting or discounted weight strategy is only naively applied. This suggests that our results (and the detailed algorithmic design & analysis) are highly non-trivial. In fact, this is also the situation in many other areas of RL theory research recently (e.g., in zero-sum Markov games), where the authors use “seemingly” the same UCB Q-learning framework to achieve more exciting results by carefully designing the algorithmic details and conducting sharper analysis.
>
> - Finally, we would like to mention a couple of other originalities of our work. In the revision, we have successfully removed the assumption about knowledge of the variation budget in each epoch, without any degradation on the regret bound. To the best of our knowledge, this is a novel result that has not been achieved in non-stationary RL with restarts, and the only existing such algorithm (Zhou et al., 2020) suffers a severe regret degeneration after removing this assumption. The revised algorithm also no longer depends on the *extra optimism*, a technique commonly used to handle non-stationary in bandit and RL problems as correctly pointed out by the reviewer. This should sufficiently distinguish our method from existing solutions in the literature (e.g., Cheung et al. 2020, Zhou et al., 2020).

---

### Official Review · AnonReviewer3 · 2020-10-29
**nonstationary episodic MDP**

**Rating:** 4
**Confidence:** 4

**Review:**

This paper presents an algorithm called RestartedQ-UCB for nonstationary episodic MDP. The setting studied is when the transition kernels and reward functions are both possibly changing with iterations. I think the setting is not quite novel, similar settings have appeared in several earlier papers, including Ortner et al. (2020) and Cheung et al. (2020).

The method proposed uses a simple restarting strategy on top of an existing algorithm with known stationary regret, for example, the Q-learning methods (Jin et al., 2018; Zhang et al., 2020). The idea of restarting strategy for nonstationary stochastic optimization or decision-making (even MDP, see reference below) is not new, which clearly diminishes the novelty of this paper. As far as I know,
1. stochastic optimization or online convex optimization. Omar Besbes, Yonatan Gur, and Assaf Zeevi. Non-stationary stochastic optimization. Operations Research, 2015.
2. multi-armed bandits. Omar Besbes, Yonatan Gur, and Assaf Zeevi. Stochastic multi-armed-bandit problem with nonstationary rewards. In NeurIPS, 2014. Omar Besbes, Yonatan Gur, and Assaf Zeevi. Optimal exploration–exploitation in a multi-armed bandit problem with non-stationary rewards. Stochastic Systems, 9(4):319–337, 2019.
3. stochastic linear bandits. Peng Zhao, Lijun Zhang, Yuan Jiang, and Zhi-Hua Zhou. A simple approach for non-stationary linear bandits. In AISTATS, 2020.
4.contextual bandits. Haipeng Luo, Chen-Yu Wei, Alekh Agarwal, and John Langford. Efficient contextual bandits in non-stationary worlds. In COLT, 2018.
5. MDP. Ronald Ortner, Pratik Gajane, and Peter Auer. Variational Regret Bounds for Reinforcement Learning. In UAI 2019. [NOTICE: this paper is published at UAI 2019 venue, while in the submission authors mistakenly cite as UAI 2020. This should be corrected.] A similar idea also appears for the tracking case (when the nonstationary setting reduces to the piecewise stationary) in UCRL2 paper (Jaksch et al. 2010, Section 7 Regret Bounds for Changing MDPs)

Some of the references are not well placed: "We measure the optimality of the policy π in terms of its dynamic regret (Cheung et al., 2020; Domingues et al., 2020)" I do not think the originality of dynamic regret (even for nonstationary MDPs) should be credited to these two papers.

As far as the extra optimism, this has also appeared in the recent work for studying nonstationary MDP (Cheung et al. 2020).

The main results (Theorem 1 and Theorem 2) are interesting, didn't the algorithm require the variation quantities Delta_r and Delate_p as input? I think this is not fully overcome in previous studies, even the most recent papers (Cheung et al., 2020; Domingues et al., 2020; Fei et al. 2020). The issue should be highlighted in the theorem and the setting of input parameters in the statement of algorithms, explicitly.

The lower bound for nonstationary MDP is not surprising by constructing a piecewise stationary hard instance and further taking a reduction to the lower bound for stationary MDP (Jin et al., 2018). Similar constructions have appeared in earlier studies of dynamic regret, to name a few, online convex optimization (https://arxiv.org/pdf/1810.10815.pdf), linear bandits (https://arxiv.org/pdf/1810.03024.pdf), etc.

Overall, the paper fails to provide much technical contribution and novelty to the community since most techniques and ideas were published previously, and unfortunately falls below the bar of ICLR. Please correct me if I missed non-trivial technical contributions.

The final issue is that in the introduction section the authors provide several practical intriguing applications with modeling by nonstationary RL/MDP, and also claim that "our model-free algorithm is more time- and space-efficient, flexible to use, and more compatible with the design of modern deep RL architectures" but no experiments are offered especially in a conference like ICLR, while the theoretical contributions seem to be unable to stand it alone. The authors should implement the algorithm in practical nonstationary RL tasks to show its strength, a number of references can be found at the survey https://arxiv.org/abs/1905.03970

---

> ### Author Response · Authors · 2020-11-20
> **Responses to Reviewer 3**
>
> We thank the reviewer for the helpful feedback. Our detailed responses are as follows.
>
> 1. First, we would like to refer the reviewer to the separate “General Responses” regarding an *emphasis* on the contributions of this paper (to establish the *minimax* regret bound in this setting with simple techniques such as restarting and Q-learning, not just *some* regret bound for non-stationary MDPs). We have also successfully removed a major assumption on knowledge of local variation budgets in the revised paper. The details can be found in the (new) Theorem 2 of the revised paper and “General Responses” as well.
>
> 2. Novelty and contributions: As pointed out in the related work part of our paper, we have clearly stated that the restarting strategy is not our main contribution. In fact, we were fully aware that it is a standard technique to handle non-stationarity in stochastic optimization, bandits, and MDPs, as the reviewer pointed out in the references. We hope to reiterate that our primary intention of this paper was not to propose a completely novel strategy to handle non-stationarity, but rather to investigate the lowest achievable regret in non-stationary RL, using techniques that are preferably as simple as possible. We would like to argue that using simple and standard techniques does not mean that our results/contributions are not novel. We believe that our main contribution should be the first near-optimal regret bound in non-stationary MDPs. Such a tight bound was previously not established in the literature, as evidenced by our summary in Table 1, and even in the very relevant papers (Zhou et al., 2020, Touati et al., 2020) that were posted after our ICLR submission.
>
> 3. In this revision, an additional novelty of our paper would be the removal of the local variation assumption without affecting the regret bound (please refer to “General Responses” for details). To our best knowledge, this result does not exist in the non-stationary RL literature with restarts, which exhibits the novelty of our method.
>
> 4. References: We thank the reviewer for pointing out the UAI reference mistake. It has been edited accordingly. In addition, to the best of our knowledge, the earliest definition of dynamic regret in non-stationary MDPs appeared in (Jaksch et al., 2010), although the exact phrase of “dynamic regret” was not coined there. We are more than happy to include any reference that the reviewer would suggest.
>
> 5. Extra optimism: Our revised algorithm no longer relies on the extra optimism (see Equation (1) in the revised paper). This should obviously distinguish our work from, e.g., Cheung et al. 2020.
>
> 6. Knowledge of variation budgets: Our revised algorithm no longer requires knowledge of local variation budgets. The details can be found in “General Responses” as well. To further remove the dependence on knowledge of the entire variation budgets, one can use the bandit-over-RL technique as in (Cheung et al. 2020).
>
> 7. Lower bounds: Such a lower bound for non-stationary MDPs did not exist previously in the literature, and we regard it as an important contribution to set a benchmark for future works of other researchers in this area. It is important to note that although the $T^{2/3}$ dependence on $T$ is expected, the exact dependence on $H, S$, and $A$ are unclear a priori; it is thus important to establish this result. To draw a comparison, (Jin et al., 2018) also characterized the exact lower bound for learning in stationary MDPs. Although their construction is built upon an existing hard instance (Jaksch et al., 2010), it is still considered as an important contribution because it was the first work to clearly state this lower bound.
>
> 8. Experiments: We have been working on the experiments since the rebuttal period started, and we will post the results hopefully by the end of the rebuttal period. We thank the reviewer for suggesting the survey paper.
>
> We sincerely thank the reviewer again for the detailed and insightful comments. Hope our updated version and the rebuttal have cleared all your concerns. We are also always open to further discussions on these technical details.

---

### Official Review · AnonReviewer4 · 2020-11-08
**A rigorous theoretical contribution to non-stationary RL**

**Rating:** 7
**Confidence:** 3

**Review:**

This paper proposes the first model-free RL algorithm (RestartQ-UCB) for the non-stationary episodic RL problems, where the model parameters are determined by an oblivious adversary and change with time with a certain budget on the total variation. Moreover, the authors provide a rigorous analysis of RestartQ-UCB and establish a near-optimal regret upper bound as well as the first lower bound on the dynamic regret in non-stationary RL.

The paper is a novel and rigorous theoretical contribution to non-stationary RL. Overall the paper is well-written and easy to follow. I have done a careful check of the proof of Theorem 1 (including the technical lemmas) and a high-level check of the rest of the analysis, and they all look sound. Overall I do not find any particular weakness in this paper. My only concern is that RestartQ-UCB requires the knowledge of the variation budget in each epoch. While this assumption has been considered in (Ortner et al., 2020), it would be helpful to provide more justification for this assumption.

Additional comments:

-While I could follow the proofs step by step in a mechanistic way, it would be helpful to first outline the high-level idea of the proofs at the beginning of Section 4.

-It is not totally clear to me why stages are necessary for the analysis. Could the authors provide some intuition behind this design?

-While the proofs of Theorem 1 and 2 are similar, it would be helpful to at least highlight the main differences in the main text.

-In Appendix B.2: In the second sentence of the first paragraph, shall the goal be proving $Q_h^{k,*}(s,a)\leq Q_{h}^{k+1}(s,a)$ instead?

---

> ### Author Response · Authors · 2020-11-20
> **Responses to Reviewer 4**
>
> We thank the reviewer for the appreciation of our paper and the helpful comments. Our detailed responses are as follows.
>
> 1. Regarding the assumption on local variation budget, we have successfully removed this assumption while still maintaining the same (nearly optimal) regret bound in the revised paper. We would like to refer the reviewer to the separate “General Responses” for more details on this point. We also remark that for non-stationary RL algorithms using restarts, the removal of this assumption was non-trivial and challenging previously. To the best of our best knowledge, the only existing such algorithm (Zhou et al., 2020) suffers a severe regret degeneration after removing this assumption.
>
> 2. We have included a high-level idea of the proofs and explanations of the differences between the two theorems in the revised paper. We thank the reviewer for pointing this out.
>
> 3. Intuition of “stages”: Intuitively, the “stage” in our paper mimics the design of discounting learning rates $\alpha_t$ in (Jin et al., 2018), where the authors put more weight on more recent samples collected from the environment. The design of stages is conceptually similar to a hard-threshold learning rate, where only the most recent $1/H$ fraction of samples are used to update the value function.
>
> 4. Regarding Appendix B.2: Yes, it should be $k+1$. We thank the reviewer for pointing out this typo. It has been edited accordingly.

---

### Author Response · Authors · 2020-11-20
**General Responses**

We thank all four reviewers for the constructive and valuable feedback.

We first would like to highlight that our main objective was to investigate the *tightest possible* regret bound in non-stationary RL, and pursue *minimax-optimal* regret as an ultimate goal. Our primary intention was not to propose a completely new strategy to handle non-stationarity, but rather to study the lowest achievable regret in non-stationary RL and its fundamental limits with preferably simple techniques. We believe that our main contribution should be the *first nearly-optimal regret bound* in non-stationary MDPs. Such a tight bound was previously not established in the literature, as summarized in our Table 1, and not even in the relevant papers (Zhou et al., 2020, Touati et al., 2020) that were posted after our ICLR submission. We sincerely hope that the contributions of our paper could be (re)-evaluated with this goal in mind, as it seems some unfortunate misunderstanding in our main contributions/purpose has occurred (see detailed explanations in the corresponding response to each reviewer).

As per the suggestions of the reviewers, we would also like to share a major update in the revision. Our previous method requires the knowledge of variation budgets in each epoch, which is the same assumption that has been made in (Ortner et al., 2020). In our revised paper, we are able to safely remove this assumption without any loss on the regret bound, as formally presented in the (new) Theorem 2 of the revised paper. This is achieved by a very lightweight modification to the algorithm. Specifically, we replace the Q-value update rule in Equation (∗) of Algorithm 1 with the new update rule in Equation (1). To understand why this simple modification works, notice that in (∗) we are adding exactly the same bonus term $2 b_\Delta$ to the upper confidence bounds of all $(s, a)$ pairs in the same epoch. Subtracting the same value from all optimistic Q-values simultaneously should not change the choice of actions in future steps. This modification incurs some additional complexity in the analysis, because the new “optimistic” $Q^k_h(s, a)$ values would no longer be *strict upper bounds* of the optimal $Q^k_h(s,a)$ anymore (as required by the analysis in stationary cases, see e.g., Lemma 4.3 in Jin et al., 2018), but only an “upper bound” subject to some error term of the order $b_\Delta$. To handle this, we provide a new analysis on how this error term propagates over time in the (new) Lemma 4.

We would like to remark that the removal of the dependence on local budget assumption requires *non-trivial* design and analysis of the UCB Q-learning algorithm. To the best of our knowledge, for non-stationary RL algorithms using restarts, the only existing algorithm that achieves this (Zhou et al., 2020) suffers a severe regret degeneration after removing this assumption. It is important to note that the revised algorithm no longer depends on the *extra optimism*, a technique commonly used to handle non-stationary in bandit and RL problems. This should sufficiently distinguish our method from existing solutions in the literature (e.g., Cheung et al. 2020, Zhou et al., 2020).


References:

Huozhi Zhou, Jinglin Chen, Lav R Varshney, and Ashish Jagmohan. Nonstationary reinforcement learning with linear function approximation. arXiv preprint arXiv:2010.04244, 2020.

Ahmed Touati, Pascal Vincent. Efficient Learning in Non-Stationary Linear Markov Decision Processes. arXiv preprint arXiv:2010.12870, 2020.

---

### Decision · Program_Chairs · 2021-01-07
**Final Decision**

**Decision:**

Reject

**Comment:**

The paper shows (nearly) matching upper and lower bounds on dynamic regret for non-stationary finite-horizon reinforcement learning problems. The paper studies an important problem and the results are interesting. Some reviewers are concerned that there is not enough algorithmic and theoretical innovations in light of prior results. Authors need to improve the presentation and add a more detailed discussion on related works, the novelty and the originality of the paper, and the new algorithmic and theoretical contributions. Finally, authors can improve the submission by implementing the proposed method and adding experiments.